# Airborne measurements and large-eddy simulations of small-scale Gravity Waves at the tropopause inversion layer over Scandinavia

Sonja Gisinger[1], Johannes Wagner[1], and Benjamin Witschas[1]

[1]Deutsches Zentrum für Luft- und Raumfahrt, Institut für Physik der Atmosphäre, Oberpfaffenhofen, Germany

**Correspondence:** Sonja Gisinger (sonja.gisinger@dlr.de)

**Abstract.** Coordinated airborne measurements were performed by the two research aircraft DLR Falcon and HALO (High Altitude and Long Range Aircraft) in Scandinavia during the GW-LCYCLE II (Investigation of the life cycle of gravity waves) campaign in 2016 to investigate gravity wave processes in the upper troposphere and lower stratosphere (UTLS) region. A mountain wave event was probed over Southern Scandinavia on 28 January 2016. The collected dataset constitutes a valuable combination of in-situ measurements and horizontal- and altitude-resolved Doppler wind lidar and water vapour measurements with the differential absorption lidar (DIAL). In-situ data at different flight altitudes and downward pointing wind lidar measurements show pronounced changes of the horizontal scales in the vertical velocity field and of the leg-averaged momentum fluxes (MF) in the UTLS region. The vertical velocity field was dominated by small horizontal scales with a decrease from around $20\,\mathrm{km}$ to $< 10\,\mathrm{km}$ in the vicinity of the tropopause inversion layer (TIL). These small scales were also found in the water vapour data and backscatter data of the DIAL. The leg-averaged MF profile determined from the wind lidar data is characterized by a pronounced kink of positive fluxes in the TIL and negative fluxes below. Largest contributions to the MF are from waves with scales $> 30\,\mathrm{km}$. The combination of the observations and idealized large-eddy simulations (LES) revealed the occurrence of interfacial waves having scales $< 10\,\mathrm{km}$ on the tropopause inversion during the mountain wave event. The contribution of the interfacial waves to the leg-averaged MF is basically zero due to the phase relationship of their horizontal and vertical velocity perturbations. Interfacial waves have already been observed on boundary-layer inversions but their concept has not been applied to tropopause inversions so far. Our idealized simulations reveal that the TIL affects the vertical trend of leg-averaged MF of mountain waves and that interfacial waves can occur also on tropopause inversions. Our analyses of the horizontal- and altitude-resolved airborne observations confirm that interfacial waves actually do occur in the TIL. As predicted by linear theory, the horizontal scale of those waves is determined by the wind and stability conditions above the inversion. They are found downstream of the main mountain peaks and their MF profile varies around zero and can clearly be distinguished from the MF profile of Kelvin-Helmholtz instability. Further, the idealized large-eddy simulations reveal that the presence of the TIL is crucial in producing this kind of trapped waves at tropopause altitude.

## 1 Introduction

Gravity waves (GWs) are an important coupling mechanism between the lower and the middle and upper atmosphere. Propagating GWs transport momentum and energy and deposit them in regions where breaking and dissipation occurs. As such,

GWs account for example for the well-known upper mesospheric wind reversals as well as the cold polar summer mesopause and the warm winter stratopause (Dunkerton, 1978; Lindzen, 1981). So far, different sources for GWs in the troposphere have been identified, e.g., flow over orography, convection, jets and fronts as well as secondary generation in the region of GW breaking (Smith, 1979; Gill, 1982; Baines, 1995; Fritts and Alexander, 2003; Sutherland, 2010; Plougonven and Zhang, 2014; Vadas et al., 2003). GWs are propagating from their sources in the troposphere and the tropopause region (Sato et al., 2009; Fritts et al., 2016). However, the atmospheric temperature and wind structures influence the propagation of GWs and alter their properties.

Starting with the work of Queney (1948) and Scorer (1949), mountain wave (MW) propagation in the atmosphere was intensively investigated using theoretical and numerical methods. An important and well known result of these investigations is that the stratospheric solution in a model taking into account a vertically varying background is not dominated by the classical solution of Queney (1948) but by reflected and downstream propagating (trapped) waves in the troposphere (Wurtele et al., 1987; Keller, 1994). The wave spectrum (i.e. wavelengths) is determined by the vertical varying wind and stability and not by the topography spectrum. The topography affects the relative amplitudes (Keller, 1994; Ralph et al., 1997). Fine scale structures in the atmosphere, such as sharp temperature inversions at the top of the boundary layer (Vosper, 2004; Sachsperger et al., 2015) or in the mesosphere (Fritts et al., 2018) can be wave guides leading to trapped waves which propagate horizontally along the inversions, i.e. interfacial waves. All those findings are in contrast to the fundamental characteristics of the hydrostatic approximation. The fundamental characteristics of the hydrostatic approximation are the absence of a mechanism which allows a wave to propagate horizontally and the consequent upward propagation of energy directly above the obstacle, regardless of the horizontal extent of the generating terrain (Wurtele et al., 1996). Linear nonrotating hydrostatic wave theory is most commonly used by MW parameterizations in weather and climate models to propagate these waves away from the subgrid-scale orography to higher levels (Eckermann et al., 2015).

Currently much activity using various ground-based, airborne, and satellite measurements is going on to get a complete picture of the GW activity and distribution around the globe and to enhance the understanding of source and propagation processes (e.g., Fritts et al., 2016; Podglajen et al., 2016; Wright et al., 2017; Shibuya et al., 2017; Kaifler et al., 2017; Krisch et al., 2017). This knowledge is required to adequately model and parameterize atmospheric GWs in weather and climate models. So far, observational indications of GW behaviour in the tropopause region such as reflection and trapping are rare due to lack of horizontal- and altitude-resolved observations in the tropopause region. Using aircraft measurements, which were taken during the Terrain-Induced Rotor Experiment (T-REX, Grubišić et al. (2008)), Smith et al. (2008) were able use in-situ data to measure partial reflection of MWs at the tropopause for the first time. Using their linear model, they identified two levels of reflection, one at the altitude where the Scorer parameter defined as

$$\ell^2(z) = \frac{N^2(z)}{U^2(z)} - \frac{\partial^2 U(z)/\partial z^2 + \partial U(z)/\partial z/H}{U(z)} - \frac{1}{4H^2},\tag{1}$$

where $N$ is Brunt-Väisälä frequency, $U$ is the cross-mountain wind speed, $z$ is altitude, and $H$ is scale height (Lane et al., 2000), changes due to changes in static stability and the other at the altitude with a discontinuity in wind speed but constant Scorer parameter.

In addition, Smith et al. (2008) and Woods and Smith (2010) found signatures of trapped waves with a horizontal wavelength of about 15 km in the in-situ measurements in the tropopause inversion layer (TIL) during T-REX. They argue that the Sierra mountain range is unlikely to be the source for those 15-km waves as such small scale waves may not reach the tropopause altitude due to the considerable evanescent decay caused by the background conditions. Instead, they suggest that those waves are generated by a nonlinear steepening process. Follow-up model simulations lead to two different explanations. First, the

short-wavelike fluctuations observed in the UTLS region are due to Kelvin-Helmholtz instability along shear lines locally induced by the primary MW, i.e. they are not trapped GWs but instead small-scale wave motions resulting from Kelvin-Helmholtz instability (Mahalov et al., 2011). Second, the downward propagating GWs, which are created by MW breaking in the middle stratosphere, and their reflection at the tropopause can create kind of lee wave trapping in the lower stratosphere (Woods and Smith, 2011).

Coordinated airborne measurements were performed by the two research aircraft DLR Falcon and HALO (High Altitude and Long Range Aircraft) in Scandinavia during the GW-LCYCLE II (Investigation of the life cycle of gravity waves) campaign in 2016 to investigate GW processes in the upper troposphere and lower stratosphere (UTLS) region. A MW event was probed over Southern Scandinavia on 28 January 2016. The collected dataset constitutes a valuable combination of in-situ measurements and horizontal- and altitude-resolved wind lidar and water vapour lidar measurements in the UTLS. In-situ

data at different flight altitudes and downward pointing Doppler wind lidar measurements revealed pronounced changes of the horizontal scales in the vertical velocity field and of the leg-averaged momentum flux (MF) in the UTLS region.

  This paper examines the MW case over Scandinavia by means of ECMWF IFS meteorological analyses and the coordinated airborne measurements of the DLR Falcon and HALO which provide horizontal- and altitude-resolved data in the UTLS. The wind data of the downward pointing Doppler lidar give the opportunity to calculate a continuous profile of MF over a two-

kilometre altitude range in the UTLS. In order to find out what determines the horizontal scales in the vertical velocity field and which process(es) can explain the observed characteristics, we investigate the possible existence of interfacial waves in the TIL (Vosper, 2004; Sachsperger et al., 2015, 2017) similar to their existence on an inversion in the troposphere (Cruette, 1976; Sachsperger et al., 2015; Chouza et al., 2015). The paper is organized as follows. Section 2 describes the models, data and methods used in this paper. Meteorological conditions and observations of the MW event on 28 January 2016 are analysed in

Section 3.1 and idealized large-eddy simulations of MW propagation in the presence of atmospheric inversions are presented in Section 3.2. The results are discussed in Section 4 and Section 5 concludes the paper.

## 2 Data and Methods

### 2.1 ECMWF global analysis

Operational analyses of the ECMWF integrated forecast system (IFS) are used to analyse the meteorological conditions on

28 January 2016. These analyses (cycle 41r1[1]) have a horizontal resolution of about 16 km on the reduced linear Gaussian

---

[1]https://www.ecmwf.int/en/forecasts/documentation-and-support/changes-ecmwf-model/cy41r1-summary-changes, last access Oct 2018

grid ($T_L$ 1279). The highest of the 137 vertical levels ( L137) is located at $\sim 80$ km (0.01 hPa). The layer thicknesses gradually increases from $\sim 300$ m at $\sim 10$-km altitude to $\sim 400$ m at $\sim 20$-km altitude and $\sim 2$ km at $\sim 60$-km altitude[2].

## 2.2 Airborne observations

### 2.2.1 Coordinated research flights on 28 January 2016

The airborne observations took place during the intensive observing period 6 (IOP 6) on 28 January 2016 in the framework of the combined missions POLSTRACC (The Polar Stratosphere in a Changing Climate), GW-LCYCLE II and SALSA (Seasonality of Air mass transport and origin in the Lowermost Stratosphere using the HALO Aircraft). An overview of the performed HALO research flights can be found in Oelhaf et al. (2019). In January 2016, the DLR research aircraft Falcon and HALO operated from the airport of Kiruna (67.82° N, 20.33° E), northern Sweden, to investigate chemical and dynamical processes in the UTLS region at high latitudes. The goal of IOP 6 was to measure a transient MW event over southern Scandinavia with coordinated cross-mountain flights of both aircraft. Figure 1a shows the operational area and the flight tracks of the research flights. The mountains were crossed at the same latitude two times by the DLR Falcon (flight legs RF07 FL2 and RF08 FL1) and three times by HALO (flight legs HL1, HL2 and HL4; Fig.1b). The limited range of the DLR Falcon required a refuel stop at Karlstad airport. The DLR Falcon was flying close to the thermal tropopause on all flight legs and measured vertical winds and GW induced momentum fluxes with the in-situ sensor at flight altitude and with the downward pointing Doppler wind lidar below the aircraft. On the flight legs HL1 and HL2 HALO was flying in the troposphere (HL1 below the DLR Falcon) and measured wave structures at flight level and in the tropopause region with the upward pointing differential absorption lidar (DIAL) measuring water vapour concentration and backscatter (WALES) (Wirth et al., 2009).

### 2.2.2 Wind lidar measurements

The DLR Falcon was among others equipped with a downward-looking coherent Doppler wind lidar (DWL) which has been operated by DLR since 1999 and which has been applied in a number of field campaigns (e.g., Chouza et al., 2017; Schäfler et al., 2018; Marksteiner et al., 2018; Lux et al., 2018). The DWL operates at a wavelength of 2 µm and is equipped with a double-wedge scanner which enables to steer the laser beam to any position within a cone angle of 30°. A more detailed description of the 2 µm DWL instrumental setup, the measurement principle, the applied retrieval algorithms and the accuracy and precision of the derived wind products is given by Chouza et al. (2015), Witschas et al. (2017), and more recently Witschas et al. (2020).

Usually, the 2 µm DWL is used to either measure the three dimensional wind vector by applying the velocity azimuth display (VAD) scan technique, or to measure vertical wind speeds by pointing the laser beam to nadir direction and compensate any attitude changes of the aircraft by means of the double wedge scanner. As shown and discussed by Witschas et al. (2017), measurements of both horizontal and vertical wind profiles are very useful to characterize the spectral properties of MWs. In order to additionally gain knowledge of the momentum transport of MWs, horizontal wind speed ($u$) and vertical wind

---

[2]https://www.ecmwf.int/en/forecasts/documentation-and-support/137-model-levels, last access Oct 2018

speed ($w$) need to be measured simultaneously. For this purpose, the 2 μm DWL has been operated with a modified scan pattern during the GW-LCYCLE II campaign for the first time. In particular, the laser beam has alternately been steered forth and back with an off-nadir angle of $\pm 20°$. With that and the knowledge of the laser beam pointing direction, $u$ and $w$ can be derived
from a successive pair of line-of-sight (LOS) measurements. It is worth mentioning that $u$ denotes the horizontal wind along flight direction here, which coincided well with the wind direction for the discussed flight (see section 3.1.1). The leg-averaged momentum flux (MF $= \overline{\rho u' w'}$) can then be calculated (Smith et al., 2016). Here, $'$ denotes perturbations of the respective quantity. In our analysis, we use spectral filters, namely Butterworth, to determine $u'$ and $w'$ for different wave classes (i.e. long, intermediate, and short waves) as it was done by Georgelin and Lott (2001). We separate the wave classes based on the
dominant horizontal wavelengths occurring in the wavelet power spectra, i.e. long ($> 30$ km), intermediate (10 km to 30 km), and short ($< 10$ km) waves (Sec. 3.1.2). For these three wave classes, the averaged MF and the corresponding uncertainty is computed. In particular, a thousand sub-legs are created as such that their start (end) point is fixed at the westernmost (easternmost) point of the measurements and the length of the leg is stepwise extended eastward (westward) by 1 km starting with a minimum length of 200 km and going up to 700 km, i.e. the full leg length. This is done to incorporate the sensitivity
of the leg-averaged MF with respect to the start/end points of the leg and the corresponding unequal sampling of updrafts and downdrafts as already suggested and analysed in a similar way by Brown (1983). We additionally found differences in the MF from DWL and HALO in-situ data at 7.8 km altitude, although one hardly can determine differences in $u$ and $w$ between DWL and HALO in-situ data. In particular, the difference in $w$ is $0.0 \pm 0.2$ ms$^{-1}$ on average. In the end, the given standard deviation accounts for these uncertainties in the MF profile being a worst case estimate with sub-legs included which have lengths shorter
than $\lambda_{\mathrm{MAX}}/2$ with theoretically $\lambda_{\mathrm{MAX}} \approx 700$ km for the discussed measurement flight.

For the applied scan pattern, each LOS measurement took 2 s and the aircraft speed above ground was approximately 200 ms$^{-1}$. Thus, the horizontal resolution of the measured horizontal and vertical wind is $\approx 800$ m. A more detailed explanation of the momentum-flux scan pattern of the 2 μm DWL will be presented in Witschas et al. (in preparation).

### 2.2.3 In-situ measurements

Horizontal and vertical velocity data at flight level are provided by the DLR facility Flight Experiments. For the DLR Falcon, the velocity field is determined from data taken by a Rosemount model 858 flow angle sensor and a Honeywell Lasernav YG 1779 inertial reference system (IRS) (Bögel and Baumann, 1991). Measurements on HALO are conducted by the Basic HALO Measurement and Sensor System (BAHAMAS). Recent method and calibration details can be found in Mallaun et al. (2015) and Giez et al. (2017). For the horizontal wind the measurement uncertainties are smaller than $0.5$ ms$^{-1}$ for HALO
and $0.9$ ms$^{-1}$ for Falcon, and smaller than $0.3$ ms$^{-1}$ for the vertical wind (Heller et al., 2017; Bramberger et al., 2018). The data are used at a time resolution of 1 s. Perturbation quantities of the velocity data ($u'$, $v'$, $w'$) for the full-leg analysis are calculated by de-trending the data with a linear least square fit and subtracting the mean over the leg (Portele et al., 2018). Wavelet spectra of vertical velocity and MF cospectra of $\rho u' w'$ (Woods and Smith, 2010) with modifications of Portele et al. (2018) are computed based on Torrence and Compo (1998)[3]. When combining the MF estimates of DWL and in-situ data,

---

[3]Wavelet software was provided by C. Torrence and G. Compo, and is available at URL: http://atoc.colorado.edu/research/wavelets/, last access Nov 2019

in-situ horizontal velocity along flight direction is used and $u'$ and $w'$ for the three wave classes are determined and analysed in the same way as described in Sec. 2.2.2.

## 2.3 Idealized numerical simulations

EULAG[4] is a multi-scale computational model for the simulation of geophysical flows. It provides at least second-order accuracy in time and space (Prusa et al., 2008). EULAG solves the governing equations of motion either in an EUlerian or a

LAGrangian form. Here, the non-hydrostatic equations of motion

$$\frac{D\mathbf{v}}{Dt} = -\nabla\frac{p'}{\bar{\rho}} + \mathbf{g}\frac{\theta}{\bar{\theta}} - \mathbf{f} \times \mathbf{v}' + \mathcal{M}', \tag{2}$$

$$\frac{D\theta}{Dt} = 0, \tag{3}$$

$$\nabla \cdot (\bar{\rho}\mathbf{v}) = 0, \tag{4}$$

are used in their Boussinesq approximated ($\bar{\rho} = \rho_0 = 1.225 \ \mathrm{kg\,m^{-3}}$) form for the first set of simulations and decrease of

density with altitude ($\bar{\rho} = \rho_0 e^{-z/H}$) is taken into account for the second set of simulations (Smolarkiewicz et al., 2001; Prusa et al., 2008). $\frac{D}{Dt}$ is the material derivative, $\mathbf{v}$ is the velocity vector, $p$ is pressure, $\rho$ is density, $\theta$ is potential temperature, $\mathcal{M}$ represents appropriate metric forces, $\mathbf{f}$ and $\mathbf{g}$ symbolize the vectors of Coriolis parameter and gravity acceleration, $z$ is altitude, and $H$ is scale height. Primes denote deviations from the ambient state and overbars refer to the horizontally homogeneous hydrostatic reference state of the Boussinesq expansion around a constant stability profile (Smolarkiewicz et al., 2001). EULAG

has been applied for a broad range of topics in fluid dynamics including orographic GWs (e.g., Prusa et al., 1996; Grubišić and Smolarkiewicz, 1997). Detailed model setup is given in Section 3.2.

# 3 Results

## 3.1 MW event over Southern Scandinavia

### 3.1.1 Meteorological situation

IOP 6 was a transient MW event over southern Scandinavia on 28 January 2016. Two synoptic low pressure systems over the tip of Greenland and over the Baltic sea caused moderate south-westerly to westerly winds ($10 \ \mathrm{m\,s^{-1}}$ to $20 \ \mathrm{m\,s^{-1}}$) in the troposphere and the excitation of MWs at the Southern Scandinavian mountain range (Fig. 2a). At tropopause level ($300 \ \mathrm{hPa}$) winds were westerly and below $30 \ \mathrm{m\,s^{-1}}$ over Southern Scandinavia as the polar front jet was located over the British Isles and northern Germany. A secondary jet streak occurred over the Norwegian Sea between Iceland and the Norwegian coast

(Fig. 2b). The vertical cross section of horizontal wind speed interpolated in time and space along flight leg RF08 FL2 shows increasing wind speed up to $80 \ \mathrm{m\,s^{-1}}$ above 20-km altitude in the stratosphere (Fig 3a). The cross section of vertical wind

---

[4]http://www2.mmm.ucar.edu/eulag/, last access Nov 2019

shows vertically propagating MWs in the troposphere and increasing wave amplitudes in the stratosphere (Fig 3b). The resolved MWs in ECWMF are associated with the main mountain peaks of the IFS model topography.

In Fig. 4, a time-height section of ECMWF horizontal wind speed located at the mountain ridge at point $X_0$ (Fig. 1a) is plotted. MWs were generated by moderate wind speeds in the lower troposphere on 28 January 2016. However, the MWs were prevented from propagating into the stratosphere until about 8 UTC due to weak winds close to $0 \, \mathrm{m \, s^{-1}}$ in the mid-troposphere. After 10 UTC, wind speeds above the tropopause and in the mid-troposphere increased which have allowed vertical propagation of tropospheric GWs into the stratosphere. During the time of the research flights (red dots in Fig. 4), wind speeds below the tropopause weakened again ($10 \, \mathrm{m \, s^{-1}}$-$15 \, \mathrm{m \, s^{-1}}$).

Vertical profiles of horizontal wind speed and Brunt-Väisälä frequency from an operational sounding[5] started from Stavanger (Fig. 1) at 12 UTC on 28 January 2016 are shown in Fig. 5a, b. This figure illustrates the moderate winds in the troposphere, the pronounced jump in static stability at the tropopause, that is typical for a TIL (Birner, 2006), and the increasing winds with height in the stratosphere within the polar vortex. The critical horizontal wavelength ($= 2\pi/\ell$) which separates evanescent and propagating GWs was mainly larger than 10 km in the troposphere (i.e. only waves with horizontal wavelength $> 10$ km can propagate), smaller than 10 km in the vicinity of the TIL, and increasing towards 20 km above in the stratosphere (Fig. 5c).

### 3.1.2 Airborne observations

The coordinated flights of HALO and DLR Falcon provided simultaneous measurements of GW induced perturbations below and in the TIL. Figure 6a shows vertical velocities observed at flight level on all 5 cross mountain flight legs. Amplitudes of $2 \, \mathrm{m \, s^{-1}}$ in the troposphere and up to $4 \, \mathrm{m \, s^{-1}}$ in the stratosphere are visible on all legs. Tropospheric measurements (HL1, HL2) show longer horizontal wavelengths compared to the observations at tropopause altitudes (RF07 FL2, RF08 FL1), which means that GW properties change in the vicinity of the tropopause. Wavelet power spectra of the observed vertical winds were computed to analyse the change in horizontal wavelengths (Fig. 6b). Wavelengths in the troposphere were in the order of 10 km to 30 km (PGS11 HL1 and HL2), while wavelet analysis shows that shorter wavelengths of 5 km to 9 km are dominating the vertical velocity of RF07 FL2 and RF08 FL1 in downstream region. Longer waves with wavelengths of $\geq 10$ km are found for the uppermost flight leg in the lower stratosphere (PGS11 HL4). Note that this was the last flight leg and it took place about two hours later than the other flight legs. The revealed wave signatures are not directly related to the topography spectrum (Fig. 6b) which was computed from the ASTER topography data (Schmugge et al., 2003) along the flight track (shown in Fig. 6a). The wave signatures are influenced by the background conditions.

Wavelet cospectra of MF were computed to study the propagation characteristics of the waves in more detail (Fig. 7). Alternating positive and negative MF at wavelengths of 10 km to 30 km were observed at distances of -100 km to 0 km and 100 km to 300 km below and in the tropopause region on the flight legs which took place at nearly the same time (PGS11 HL1 and HL2, RF08 FL1). This alternating pattern is an indication for reflected and trapped waves (Woods and Smith, 2010, see also Sec. 3.2). Significant MF at shorter wavelengths is found in the tropopause region and strongest alternating positive and negative signals occur downstream of the main mountain peaks. No significant positive or negative MF is found for the short

---

[5]sounding data from http://weather.uwyo.edu/upperair/

scales $< 10$ km above the tropopause at 13 km altitude (PGS11 HL4). These findings indicate that the short waves are trapped in the tropopause region. Upward propagating longer waves (negative MF) with horizontal wavelengths of approximately 40 km to 50 km are found for RF08 FL1 and PGS11 HL4 at 120 km distance. Positive MF for the long waves is found in RF08 FL1 at 220 km distance which could be caused by partial reflection of these waves. Based on the wavelet spectra, three wave classes can be distinguished according to their horizontal scales: long ($> 30$ km), intermediate (10 km to 30 km), and short ($< 10$ km) waves.

The DLR Falcon DWL measured in nadir mode on the first cross-mountain flight leg RF07 FL2 (Fig. 8). Measured vertical winds show fine scale up- and downdrafts over the mountains. The horizontal wavelengths of the GWs are smaller downstream of $X_0$ on the lee side of the mountain and the phase lines are vertical. This again indicates wave trapping.

Lidar and in-situ measurements of the coordinated flight legs RF08 FL1 and PGS11 HL1 are shown in Fig. 9. As the DWL operated in flux mode on this leg, both the horizontal wind component in flight direction and the vertical wind component were measured. Contour lines of lidar measurements are overlaid by in-situ wind measurements of both aircraft. Wind measurements are complemented by water vapour measurements of the upward looking DIAL aboard HALO. Horizontal wind speeds in Fig. 9a show large scale wave structures with upstream tilted phase lines in the troposphere. In-situ measurements around the tropopause indicate similar wave structures but with stronger wind speeds compared to tropospheric values. In addition, large-scale wave structures are superimposed by small-scale waves with vertical phase lines. These small scale-waves are more clearly visible in vertical wind measurements, which are more sensitive to smaller scale waves (Lane et al., 2003; Smith and Kruse, 2017), and show a clear change from intermediate to smaller wavelengths below and in the vicinity of the tropopause (Fig. 8 and 9b). This behaviour was already revealed by the wavelet analysis of the in-situ measurements (Fig. 6b). The short horizontal wavelengths are also visible in observations of water vapour mixing ratio (Fig. 9c) and the lidar backscatter ratio at 1064 nm (Fig. 9d) between 8 km and 10 km altitude. A backscatter ratio $> 1$ in the DIAL data reveals the presence of aerosols and clouds.

Additional information regarding the wave propagation comes with the direct measurements of GW induced momentum fluxes by the new momentum flux method of the Falcon DWL. This was done on the coordinated flight leg RF08 FL1. Figure 10 illustrates vertical profiles of leg-averaged momentum fluxes along the cross-mountain flight legs RF08 FL1, PGS11 HL1 and HL4 obtained from lidar and in-situ measurements. The MF profiles can be distinguished for the three wave classes defined above. The most prominent feature is the kink reaching positive values for the long waves between 8-km and 9-km altitude. Negative fluxes of the same magnitude are found below. This strengthens the previous assumption that waves are partially reflected at the TIL. The mean MF at 7.8-km altitude of the DWL and the HALO in-situ data differs but within the range of uncertainty. It is worth mentioning that the uncertainty in the MF from the in-situ data is largest at this altitude and larger than the uncertainty derived for the DWL data. This means the MF from in-situ at this altitude could be biased to MF of larger magnitude due to localized peaks in $\rho u'w'$ along the leg. The intermediate and short scale waves show similar MF profiles with small undulations around zero. The leg-averaged momentum flux of the long waves is positive (around $-0.05$ Pa) at 13.3-km altitude which could be a hint for wave reflection in the stratosphere or a stratospheric source creating downward propagating GWs.

## 3.2 Idealized simulations of MWs and the TIL

In this section, it is investigated if interfacial waves on an inversion found in the lower troposphere can also occur at tropopause altitudes and which conditions are necessary for their occurrence. It is tested if this wave trapping was possible on 28 January 2016 over southern Scandinavia by performing additional 2-dimensional simulations with the Scandinavian topography and background profiles which approximate the prevailing conditions on that day. There is no intention to tune the simulations as close to the measurements as possible because the main goal in this study is to identify processes which could explain the observed wave structure in the UTLS.

The computational parameters are chosen similar to Vosper (2004) for the first set of simulations. The 2-dimensional domain consists of 1032 and 2000 grid points in $x$- and $z$-direction, respectively, with grid increments of $\Delta x = 100$ m and $\Delta z = 10$ m (terrain following). This results in a total domain size of about $103\,\text{km} \times 20\,\text{km}$. The integration time step $\Delta t$ is set to 1 s. Open boundaries are applied in $x$-direction. The model top is a rigid lid. The sponge layers at the horizontal edges of the domain are 8 km wide and the sponge layer at the top of the domain starts at 15-km altitude. As in Vosper (2004) an idealized ridge

$$
h(x) = \begin{cases} h_0[1 + \cos(Kx)]/2 & \text{for } |x| \leq \pi/K \\ 0 & \text{for } |x| > \pi/K, \end{cases} \tag{5}
$$

where $K = 2\pi/L$, and a free slip lower boundary condition is used. Mountain height $h_0$ is set to 400 m and width $L$ to 10 km or 5 km. The usage of this idealized ridge is considered to be sufficient to investigate the occurrence and changes in the horizontal scale of the GWs in the vicinity of the TIL because it is known that the horizontal wavelength of interfacial waves is independent of the mountain half-width and height (Sachsperger et al., 2017). Four simulations are performed with a vertically constant horizontal velocity $U = 8\,\text{m}\,\text{s}^{-1}$ and different profiles of potential temperature with the corresponding Brunt-Väisälä frequency as initial conditions. The integration time step $\Delta t$ is set to 1 s. The total integration time for these simulations is between 96 and 190 minutes. The initial disturbance created by the mountain during the initialization of the simulations has moved far enough downstream in the region of interest and the simulations have reached quasi steady state by that time. Table 1 summarizes the relevant initial parameters and total integration time for the different model runs.

Figure 11a, b (RUN 1 and 2) shows that the EULAG model can reproduce the findings of Vosper (2004): an inversion at the top of an neutral boundary layer leads to downstream propagating waves at the altitude of the inversion (Fig. 11b). The horizontal wavelength is approximately 5 km and the largest amplitudes are found in the vicinity of the inversion (Fig. 12c). The signal downstream of the terrain is weak below the inversion (Fig. 12d) and absent if no inversion is present (Fig. 12a, b). If the stability above the inversion is increased to $N_U = 0.02\,\text{s}^{-1}$ (stratospheric stability), the strength of the inversion $\Delta\theta$ must be twice as large to allow for wave trapping and horizontal propagation on the inversion (RUN 3). The horizontal wavelength of the interfacial waves decreases with increasing stability above the inversion. The horizontal wavelength is approximately 2.5 km for $U = 8\,\text{m}\,\text{s}^{-1}$ and $N_U = 0.02\,\text{s}^{-1}$ (Fig. 12f). This is because for $N_U = 0.02\,\text{s}^{-1}$ waves with a horizontal wavelength of 5 km are no longer evanescent above the inversion and can propagate vertically (no trapping).

For Run 2, wave classes can be defined according to their horizontal scales, i.e. $> 6 \, \text{km}$ for intermediate MWs and $< 6 \, \text{km}$ for short waves. Figure 13 shows the MF (profiles) for the two wave classes. In Figure 13b, it can be seen that the short wave class not only contains interfacial waves but also some upward propagating non-hydrostatic MWs that are close to the mountain at low levels and propagating further downstream at higher levels. When MF is averaged for the whole domain, the MF profiles

of the intermediate and the short waves show a distinctive kink at the altitude of the inversion (Fig. 13e). This is not found when the inversion is absent (Fig. 13e). When MF is averaged for the downstream region, the resulting MF of the interfacial waves depends on the exact start/end points of the downstream region with respect to the wave phase. This is because the interfacial waves show alternating positive and negative fluxes downstream of the mountain (Fig. 13b). In contrast to upward propagating mountain waves (Fig. 13d), the phase shift between u' and w' is -90° for interfacial waves below the inversion

(Fig. 13c) and changes to +90° right above the inversion (not shown). When MF is averaged over a downstream region that only contains full wave cycles, the resulting MF is zero for well-established and totally trapped interfacial waves (Fig. 13e). When the start/endpoints of the downstream region are chosen such that waves are partly included, MF can be neg. (pos.) right below the inversion and positive (negative) above the inversion. The sign of MF depends on the cutting location in the wave cycles, i.e. depends on if more negative (positive) MF is included in the average (Fig. 13b). This MF profile of the interfacial waves

differs from the MF profile of Kelvin-Helmholtz instability for which MF is zero below and above the instability (Mahalov et al., 2011).

Figures 11d and 12h (RUN 4) show that interfacial waves can also exist in the TIL located between a stably stratified troposphere ($N_L = 0.01 \, \text{s}^{-1}$) and the stratosphere ($N_U = 0.02 \, \text{s}^{-1}$). The horizontal wavelength is again approximately $2.5 \, \text{km}$ for $U = 8 \, \text{m} \, \text{s}^{-1}$. Besides the upward propagating MWs and the interfacial waves, reflected waves with a horizontal wavelength

of approximately $5 \, \text{km}$ exist downstream of the mountain in the troposphere (Figs. 11d and 12j) although the classical trapping condition of a decreasing Scorer parameter with altitude in the troposphere (Scorer, 1949) is not fulfilled. The amplitudes of the reflected waves in the troposphere are found to be larger if an inversion is present at the tropopause than if there is just the jump from tropospheric to stratospheric stability (not shown). The results of these numerical simulations confirm that interfacial waves can exist in the TIL. The mechanism of their trapping (evanescence in the layer above the inversion), the resulting

horizontal wavelength, and the occurring horizontal propagation match the published results for boundary-layer inversions (Vosper, 2004; Sachsperger et al., 2015).

The second set of simulations (RUN 5 and 6) uses a 2-dimensional domain with 2016 and 1000 grid points in $x$- and $z$-direction, respectively, with grid increments of $\Delta x = 500 \, \text{m}$ and $\Delta z = 40 \, \text{m}$. This results in a total domain size of about $1008 \, \text{km} \times 40 \, \text{km}$. The total integration time for these simulations is 16 hours. This is longer than for the other set of sim-

ulations because it takes longer for the initial disturbance to reach the border of the larger domain. In contrast to the single mountain simulations, these simulations having more complex topography do not reach quasi steady state due to continuous interaction of waves from the different mountain peaks, reflected and trapped waves in the troposphere, and interfacial waves. Open boundaries are applied in $x$-direction. The model top is a rigid lid. The sponge layers at the horizontal edges of the domain are 40 km wide and the sponge layer at the top of the domain starts at 25 km altitude. The Scandinavian topography

is interpolated on the 500 m grid from ASTER data. The initial profiles approximate the background conditions over Southern

Scandinavia given by the Stavanger radiosonde on 28 January 2016 (Fig. 14a, b, d, e). Simulations without and with a TIL are performed. The simplified initial horizontal velocity profile does not contain negative shear above the tropopause but negative shear establishes in the course of the simulation (black dashed profiles shown in Fig. 14a, d are located at -150 km distance (Fig. 15f) 16 hours after start of the simulations).

Figures 14f and 15h (RUN 6) show that interfacial waves can also exist for the background conditions found on 28 January 2016 over Southern Scandinavia. They are found downstream of the main mountain peak in the vicinity of the TIL (Fig. 14f and Fig. 15f) and their horizontal wavelength is approximately 8 km (Fig. 15h). The horizontal band pattern in the wavelet spectrum resembles the one of the idealized mountain simulations presented above (Fig. 12c, f, h). Small scale interfacial waves are absent in the case of no TIL (Fig. 14e and Fig. 15a, c). They are only found in the TIL (Fig. 15f, h) but not below
(Fig. 15g, i). Reflected waves with horizontal wavelengths between 10 km and 30 km exist downstream of the main mountain peaks in the troposphere (Figs. 14e, f and 15d, i). It was already mentioned that the horizontal wavelength of the interfacial waves is independent of the generating terrain and determined by the background wind and stability. These 2-dimensional simulations reveal the expected wavelength of the GWs over Southern Scandinavia downstream of the main mountain ridge on 28 January 2016, i.e. approximately 8 km in the vicinity of the TIL and between 10 km and 30 km in the troposphere.
However, the interfacial waves in the simulation are not as dominant as in the measurements (Fig. 15h vs. RF07 FL2 and RF08 FL1 in Fig. 6b). There is a stronger signal of the upward propagating MWs above the main mountain peaks (Fig. 15f, h).

MF profiles for the three wave classes (long ($> 30$ km), intermediate (10 km to 30 km), and short ($< 10$ km)) are computed in the same way as for the measurements (Sec. 2.2.2) and are presented in Figure 16. The MF profiles of the three wave classes clearly distinguish from each other. The fact that the mean MF profile computed from the set of sub-legs is close to the MF
profile averaged for the full leg distance, which has the largest likelihood to capture the full wave cycles of the wave packages, supports that the sub-legs are chosen in a proper way. The pronounced kink in the MF profiles of the long and the short waves in the altitude range of 7 km to 9 km is a clear feature of the effect of the TIL (Fig. 16b) and not visible in the no-TIL simulation (Fig. 16a). The amplitudes of the long waves (i.e. MWs) and the intermediate waves (i.e. reflected and trapped waves in the troposphere) and their resulting MF are overestimated compared to the observations (Fig. 6 and Fig. 15f-i). The
MF is overall negative for these simulations but close to zero for the short waves. These findings are most likely an effect of the 2-dimensional model setup. Interestingly, the MF of the long waves shows a larger magnitude in the simulation without TIL (Fig. 16a) compared to the simulation with TIL (Fig. 16b). The MF of the intermediate waves shows an opposite behaviour, i.e. smaller in magnitude in the simulation without TIL. This change in MF between the two simulations suggests stronger reflection of the MWs at the TIL. This is a finding that is in agreement with findings from the single mountain simulations.

**4   Discussion**

The atmospheric conditions during the MW case were characterized by moderate low level flow ($\sim 10\,\mathrm{m\,s^{-1}}$), comparatively weak wind speed ($\sim 30\,\mathrm{m\,s^{-1}}$) around the TIL and increasing wind speed above (Sec. 3.1.1). The coordinated airborne measurements including the downward pointing Doppler wind lidar measurements revealed that the vertical velocity field was

dominated by small horizontal scales with a decrease from around 20 km to < 10 km in the vicinity of the TIL. These small
scales were also found in the water vapour data and backscatter data of the DIAL (Sec. 3.1.2). The corresponding MF indicates
wave reflection and trapping at the TIL (Fig. 7 and Fig. 10). It is known that atmospheric inversions can be wave guides leading
to wave trapping and downstream wave propagation (Vosper, 2004; Sachsperger et al., 2015; Chouza et al., 2015; Fritts et al.,
2018) but observations of downstream wave propagation of small scale waves at tropopause inversions are rare (Smith et al.,
2008; Woods and Smith, 2010).

In the course of investigating MW propagation in the UTLS with horizontal- and altitude-resolved airborne observations
(Section 3.1.2), we found that the measured and simulated MF profile of the short waves (< 10 km) does not match the typical
profile of Kelvin-Helmholtz instability that is characterised by one peak of positive MF (Mahalov et al., 2011). The shear was
not strong enough to generate Kelvin-Helmholtz instabilities (Fig. 9a and Fig. 14a). Instead, the MF profile of the short waves
varies around zero (Fig. 10). Trapped waves in the troposphere are known to have leg-averaged MFs of around zero (Woods
and Smith, 2010; Georgelin and Lott, 2001). Our analyses revealed that the same is true for interfacial waves propagating
horizontally along inversions. The MF profile of the long waves (> 30 km) is characterized by negative fluxes below and
positive fluxes in the TIL which show similar magnitudes (Fig. 10). This is most likely due to partial reflection of these waves
at the TIL. In the lower stratosphere, the leg-averaged MF was found to be positive (around 0.05 Pa). This is in contrast to
the findings during DEEPWAVE where no positive leg-averaged MF was found in the lower stratosphere above New Zealand
([Fig. 5b in (Smith et al., 2016)]. However, analyses in Smith et al. (2016) are limited to waves having scales < 150 km and
at least in ground-based lidar data downward propagating waves were frequently observed in wintertime in the stratosphere
above New Zealand (Kaifler et al., 2017). Local values of MF cospectra reach up to 0.034 $\mathrm{kN\,m^{-1}}$ in magnitude which is
slightly below or half as large as the values found for an MW event during DEEPWAVE which were between 0.05 $\mathrm{kN\,m^{-1}}$
and 0.07 $\mathrm{kN\,m^{-1}}$ in magnitude (Portele et al., 2018).

The observed horizontal wavelengths in the vicinity of the TIL were clearly evanescent in the stratosphere (Fig. 5c). This
excludes their direct propagation from above followed by their trapping in the TIL similar to Woods and Smith (2011). The
presence of interfacial waves that are trapped on inversions and that are likely generated by MWs coming across the inversions
(Sachsperger et al., 2017) has not yet been observed at TILs. Linear theory is able to describe the horizontal wavelength and
the propagation of the interfacial waves (Vosper, 2004; Sachsperger et al., 2015, 2017). However, the amplitudes depend on
the energy source, which is better described by hydraulic theory than by traditional linear models (Sachsperger et al., 2017).
The traditional linear models link the energy source to the topography which is inaccurate for interfacial waves (especially for
large amplitudes) because of nonlinear effects. Sachsperger et al. (2017) included the nonlinear effects in their model for the
amplitudes of interfacial waves on the boundary-layer inversion by assuming that the interfacial waves originate at the density
interface further aloft in the interior of the fluid and the energy source for these non-hydrostatic lee wave train is the energy
convergence at an internal jump between two fluid layers of different densities.

The determined wave properties (observations) match those of interfacial waves (simulations) for which stability and wind
conditions above the inversion determine the horizontal scales of the waves (Vosper, 2004; Sachsperger et al., 2015). The
performed simulations show that the presence of the TIL is crucial in producing the trapped waves at tropopause altitude and

vertical wind shear by the main MW was not sufficient in this case (Fig. 14a). However, the amplitudes of the interfacial
waves were underestimated compared to the long and intermediate waves in the simulations. This can have several reasons.
The amplitudes of interfacial waves depend on the amount of energy provided by the main wave source at the interface
and the acting nonlinear processes (Sachsperger et al., 2017). It was not yet investigated how the interaction and generation
processes depend on the model resolution and if the amplitudes increase with increasing resolution. The simulations were only
2-dimensional so they cannot capture effects of the fully 3-dimensional mountain range. Moreover, potential additional energy
input by downward propagating larger scale waves from stratospheric sources (e.g., polar night jet (Dörnbrack et al., 2018)) or
reflection of MWs in the mid and upper stratosphere are not included in the simulations. The positive leg-averaged MF for the
long waves computed from HALO in-situ data at an altitude oft 13.3 km could be a hint for such additional energy input. In
addition, also Krisch et al. (2020) found for the same HALO flight a chequerboard pattern in their 3-dimensional temperature
observations that allow to study large scale waves ($\mathcal{O}(100$ km$)$) below the aircraft in the upper troposphere and suggest a
possible level of reflection above flight altitude. The evaluation of these effects and the assessment of their sensitivities require
additional extensive model simulations which are beyond the scope of this paper and can be addressed in a future study.

## 5   Conclusions

The unique combination of observations from coordinated airborne in-situ and lidar measurements and idealized large-eddy
simulations revealed the occurrence of interfacial waves on the tropopause inversion during an MW event over Southern
Scandinavia on 28 January 2016. Such interfacial waves have already been observed on boundary-layer inversions but their
concept has not been applied to tropopause inversions so far.

Strong shear induced by the main MW can cause Kelvin-Helmholtz instabilities which results in similar patterns in the
vertical velocity field (Mahalov et al., 2011). Although the horizontal scales close to ten kilometres, which is similar to T-REX
observations (Smith et al., 2008; Woods and Smith, 2010), neither Kelvin-Helmholtz instability nor downward propagation of
small scale secondary GWs generated by MW breaking in the middle stratosphere can explain our observations. The vertical
shear was not as pronounced as during the T-REX case (Mahalov et al., 2011) because the tropopause jet was not well estab-
lished over Southern Scandinavia on 28 January 2016. The wind speed influenced by the large scale MW was only between
$10 \, \mathrm{m \, s^{-1}}$ and $40 \, \mathrm{m \, s^{-1}}$. The stratospheric critical horizontal wavelengths calculated from co-located radiosonde measurements
are larger than the observed scales in the UTLS region which would hinder their direct downward propagation from a breaking
region located higher up.

Our idealized simulations reveal that interfacial waves can occur also on tropopause inversions similar to boundary layer
inversions. Our analyses of the horizontal- and altitude-resolved airborne observations confirm that they actually do occur. As
predicted by linear theory, the horizontal scale of those waves is determined by the wind and stability conditions above the
inversion. They are found downstream of the main mountain peaks and their MF profile varies around zero. That is similar to
tropospheric trapped waves and clearly distinguishes from the MF profile of Kelvin-Helmholtz instability.

*Data availability.* HALO and Falcon data is available on HALO-Database (https://halo-db.pa.op.dlr.de/mission/3). ECMWF data is available in the MARS archive (https://apps.ecmwf.int/archive-catalogue/). EULAG simulation data is available on the ftp-server (ftp://ftp.pa.op.dlr.de/pub/gisinger/ACP_EULAG_2020/).

*Author contributions.* S. Gisinger performed the large-eddy simulations and prepared the manuscript with contributions from all co-authors.
420  J. Wagner did the analyses of the meteorological conditions and the aircraft in-situ data. B. Witschas did the analyses of the lidar data.

*Competing interests.* The authors declare that they have no conflict of interest.

*Acknowledgements.* This research was funded by the German research initiative "Role of the Middle Atmosphere in Climate (ROMIC/01LG1206A)" funded by the German Ministry of Research and Education in the project "Investigation of the life cycle of gravity waves (GW-LCYCLE)" and by the Deutsche Forschungsgemeinschaft (DFG) via the Project MS-GWaves (GW TP/DO 1020/9 1). We thank the PIs and all partic-
ipants of the GW-LCYCLE II campaign, in particular, Prof. Markus Rapp and Dr. Andreas Dörnbrack for input and discussions. We thank Dr. Stephan Rahm and Dr. Martin Wirth for providing the lidar data and Martina Bramberger and Tanja Portele for sharing their code and knowledge for the in-situ data analyses. Access to the ECMWF data was possible through the special project "HALO Mission Support System". Computational resources for EULAG simulations were provided by DKRZ. Radiosonde data were downloaded form the public archive of the University of Wyoming.

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

580

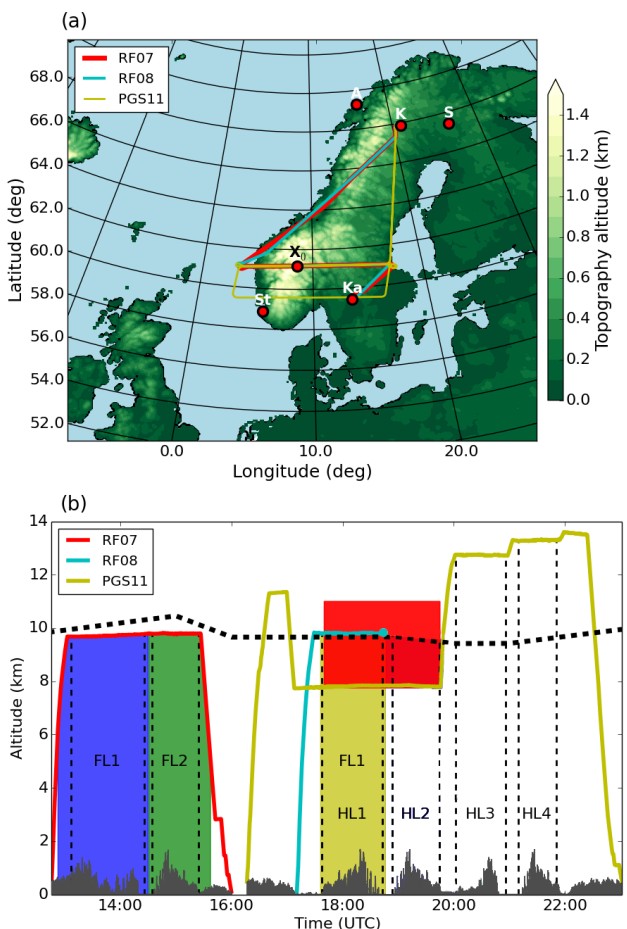

**Figure 1.** (a) Topographic map of Scandinavia and area of operation of IOP 6 during the GW-LCYCLE II campaign. The coloured lines indicate Falcon (RF07 and RF08) and HALO (PGS11) flight tracks. The red dots mark the position of Andenes (A), Kiruna (K), Sodankylä (S), Karlstad (Ka) and Stavanger (St). The location of the highest mountain peak on the cross-mountain flight legs is marked with $X_0$. Flight altitudes of Falcon and HALO are shown in (b). Falcon flight legs RF07 FL2 and RF08 FL1 and HALO flight legs PGS11 HL1, HL2 and HL4 are cross-mountain flights through $X_0$, which are analysed in this study. Colour shaded areas mark regions covered by the upward looking HALO water vapour lidar WALES (red) and the downward looking Falcon Doppler wind lidar in scanning (blue), nadir (green) and flux mode (yellow). The temporal evolution of the ECMWF thermal tropopause height at point $X_0$ is indicated with the thick dashed line.

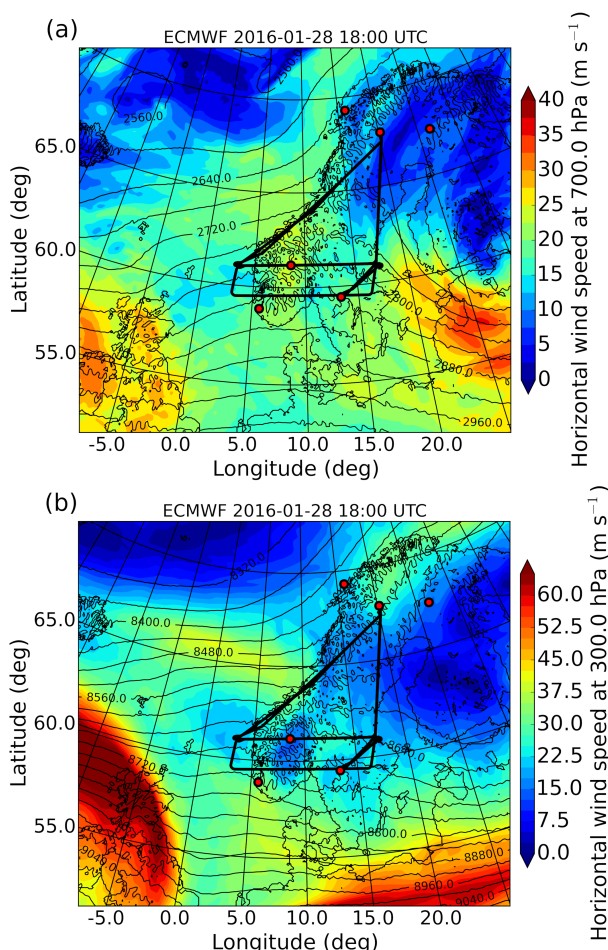

**Figure 2.** Meteorological maps of horizontal wind speed and geopotential height (black contour lines) at (a) 700 hPa and (b) 300 hPa at 18 UTC on 28 January 2016 obtained from the ECMWF model. Black lines indicate flight legs of the three research flights and red dots mark the same locations as in Figure 1.

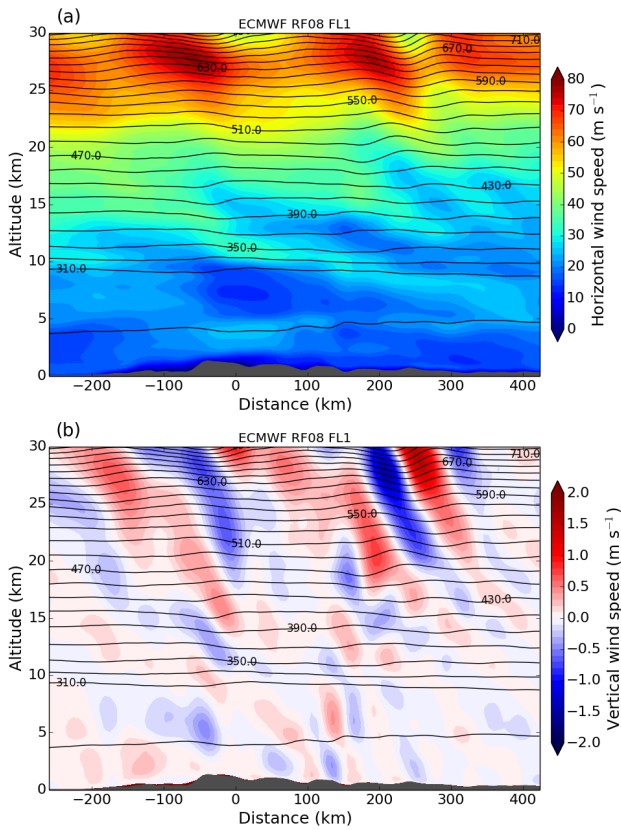

**Figure 3.** ECMWF vertical cross sections of (a) horizontal wind speed and (b) vertical wind speed interpolated in time and horizontal space along flight leg RF08 FL1. Black contour lines indicate potential temperature with an interval of 20 K. The cross section distance is centered at $X_0$ (see Fig. 1).

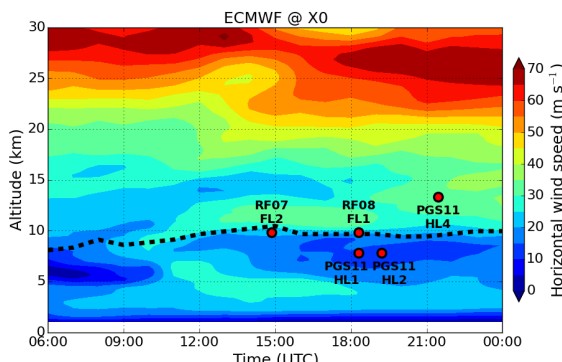

**Figure 4.** ECMWF time-height section of horizontal wind speed at point $X_0$. The black dashed line marks the height of the thermal tropopause. Red dots mark the altitudes of HALO and Falcon at point $X_0$ of the respective flight legs (see also Fig. 1).

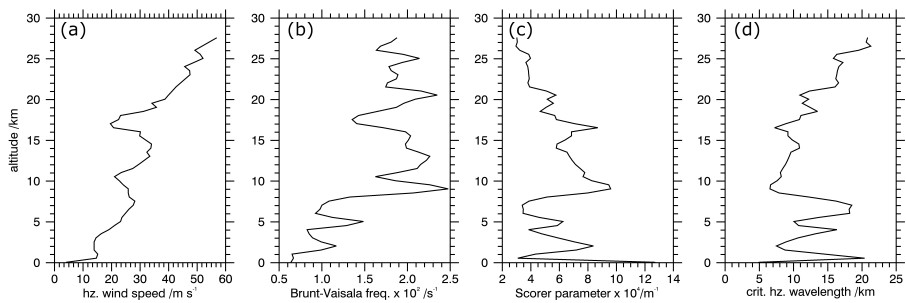

**Figure 5.** Vertical profiles of (a) horizontal wind speed, (b) Brunt-Väisälä frequency, and (c) critical horizontal wavelength of the radiosonde launched at Stavanger (St) southern Norway (Fig. 1) at 12 UTC on 28 January.

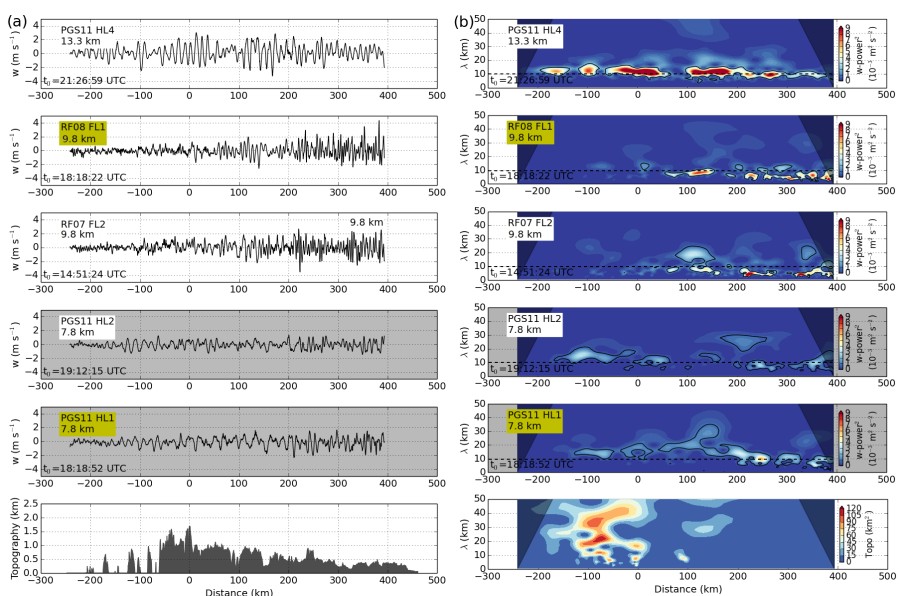

**Figure 6.** Cross mountain flight legs of Falcon and HALO for (a) in-situ vertical wind and topography and (b) corresponding wavelets with the horizontal wavelength given on the y-axis. The dashed horizontal line marks 10-km horizontal wavelength that separates short and intermediate scale waves. Black contour lines mark regions significant at the 95%-confidence level. The cone of influence is shaded in grey. Flight legs located below the tropopause (see labelled mean flight altitudes) are marked with grey background colour. Time $t_0$ indicates when the aircraft was located at $X_0$ (see Fig. 1) and shows that PGS11 HL1 and RF08 FL1 (labelled with yellow boxes) took place nearly at the same time (HALO was flying 30 seconds behind Falcon).

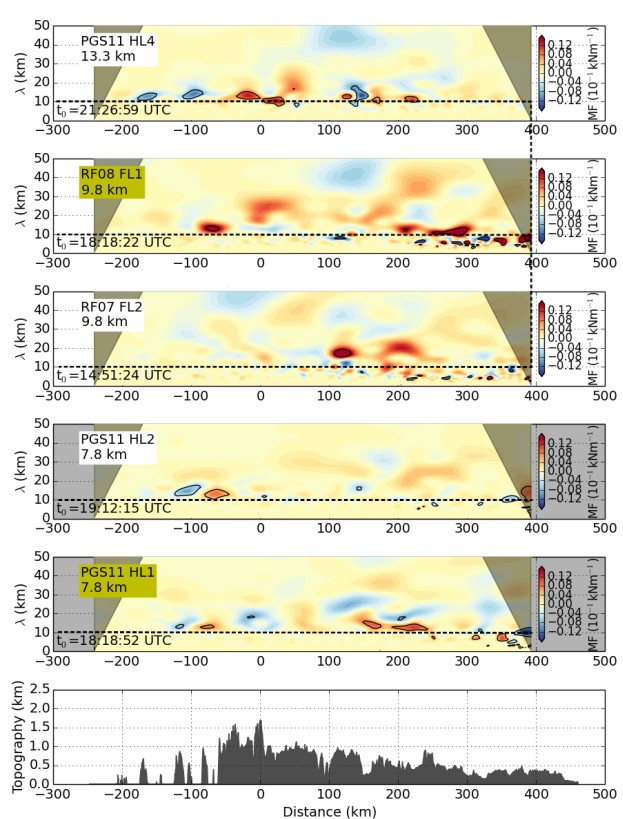

**Figure 7.** As in Fig. 6b but showing wavelet cospectra of MF ($\rho u'w'$).

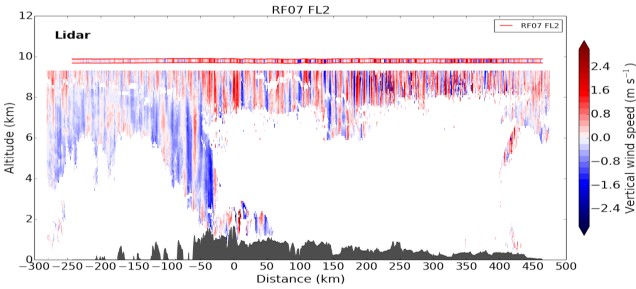

**Figure 8.** Vertical winds along flight leg RF07 FL2 measured by the DWL and in-situ instruments (marked by red horizontal lines) at flight level by the DLR Falcon.

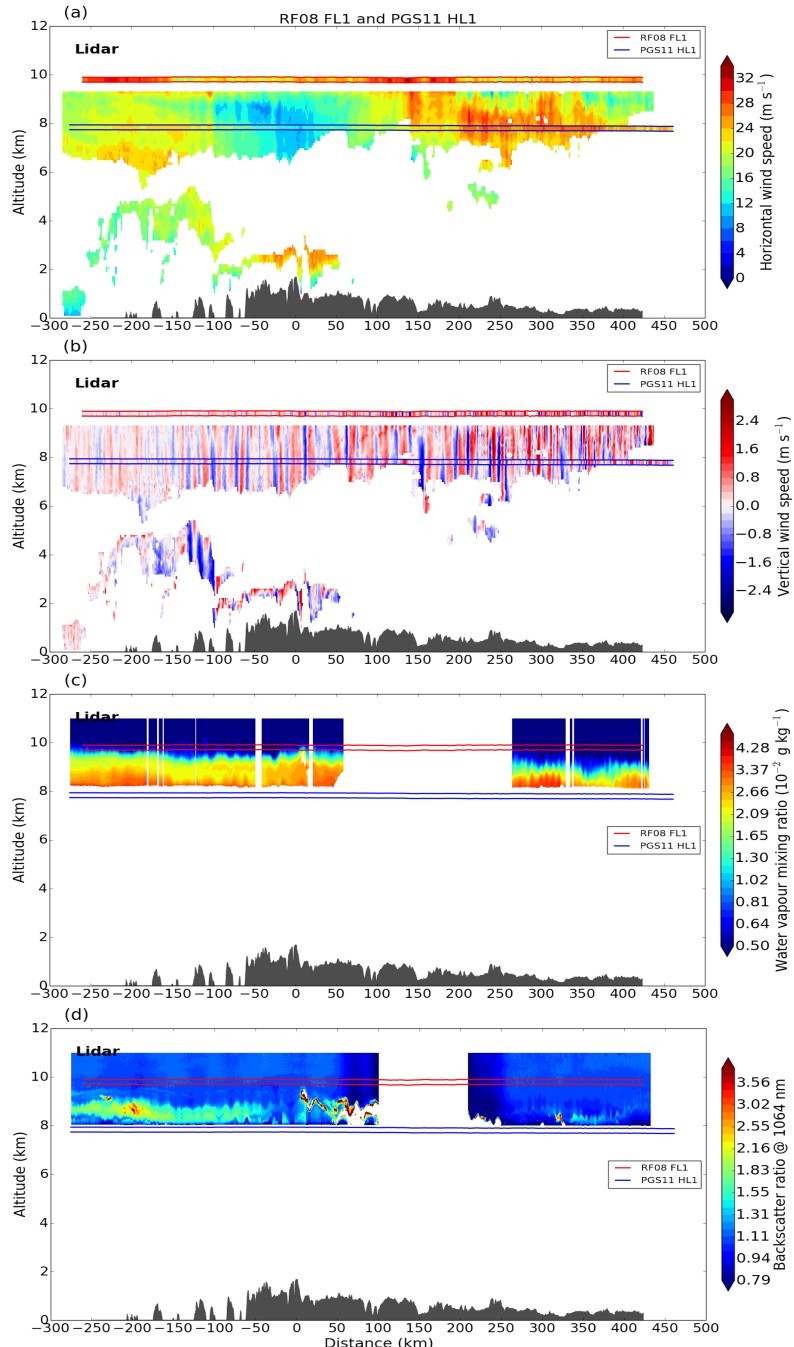

**Figure 9.** DWL measurements of (a) horizontal wind speed and (b) vertical wind speed and WALES measurements of (c) water vapour mixing ratio and (d) lidar reflectivity along flight leg RF08 FL1/PGS11 HL1 combined with corresponding in-situ measurements of HALO and DLR Falcon at flight level (marked by blue and red horizontal lines).

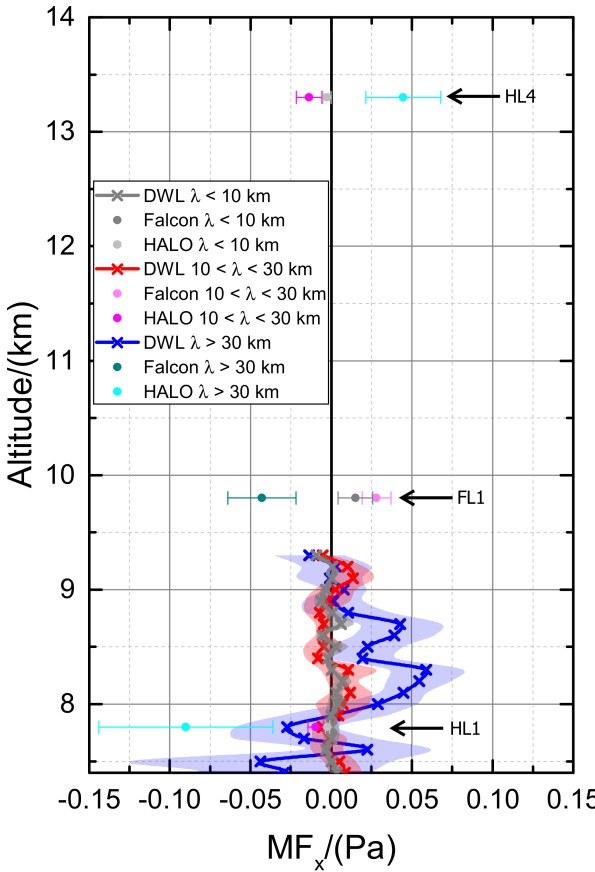

**Figure 10.** Leg-averaged momentum fluxes as mean for varying leg length (solid) with standard deviation (shading, error bars) along flight leg RF08 FL1 obtained from DWL (×) and in-situ measurements (●) which include also PGS11 HL1 and HL4. Three wave classes are colour coded and bold black line separates positive and negative fluxes.

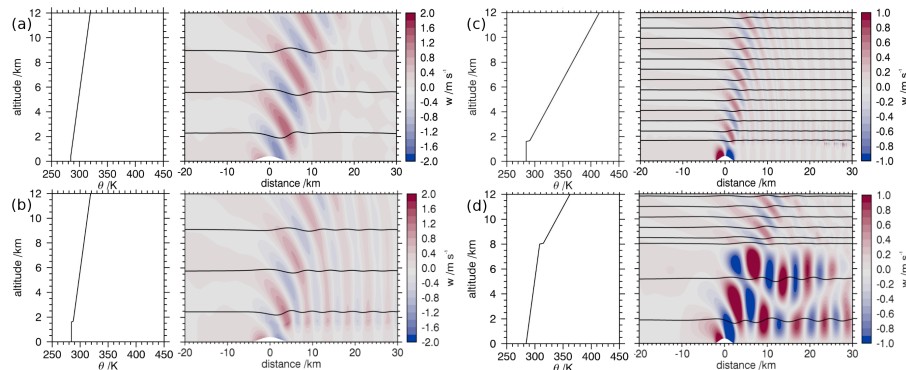

**Figure 11.** Potential temperature and vertical velocity of the idealized simulations of the cases with (a) a neutral boundary layer without inversion and $N_U = 0.01$ s$^{-1}$ (RUN 1), (b) a neutral boundary layer with an inversion of 3.3 K and $N_U = 0.01$ s$^{-1}$ (RUN 2), (c) a neutral boundary layer with an inversion of 6.6 K and $N_U = 0.02$ s$^{-1}$ (RUN 3), and (d) a stable troposphere ($N = 0.01$ s$^{-1}$) with a TIL of 6.6 K and $N_U = 0.02$ s$^{-1}$ (RUN 4).

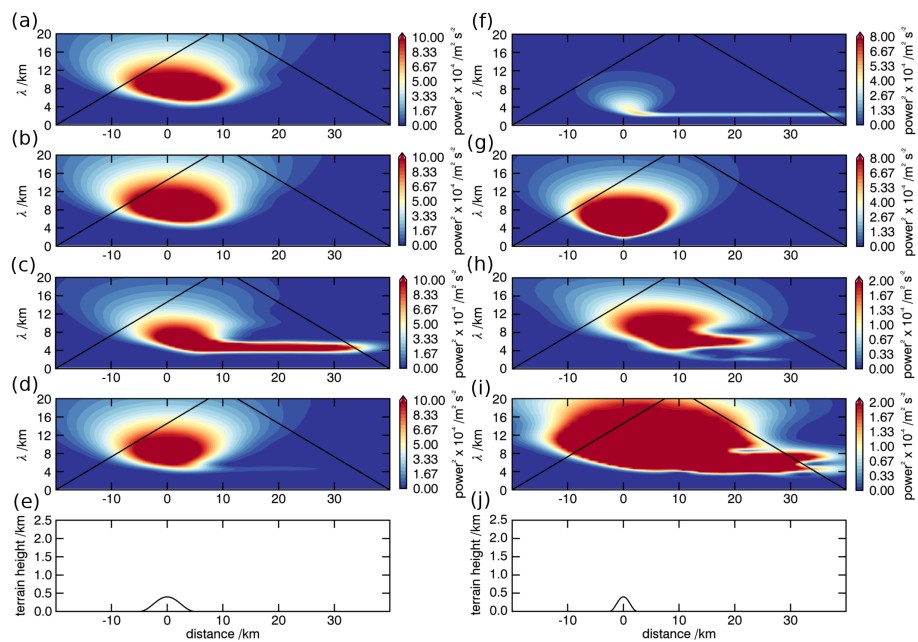

**Figure 12.** Wavelet spectra of the vertical velocity of the idealized simulations shown in Fig. 11 in the vicinity of (a, c, f, h) and below (b, d, g, i) the top of the boundary layer or the inversion layer. e and j show the idealized terrain. (a, b) is for a neutral boundary layer without inversion and $N_U = 0.01\ \mathrm{s}^{-1}$ (RUN 1), (c, d) for a neutral boundary layer with an inversion of 3.3 K and $N_U = 0.01\ \mathrm{s}^{-1}$ (RUN 2), (f, g) for a neutral boundary layer with an inversion of 6.6 K and $N_U = 0.02\ \mathrm{s}^{-1}$ (RUN 3), and (h, i) for a stable troposphere ($N = 0.01\ \mathrm{s}^{-1}$) with a TIL of 6.6 K and $N_U = 0.02\ \mathrm{s}^{-1}$ (RUN 4).

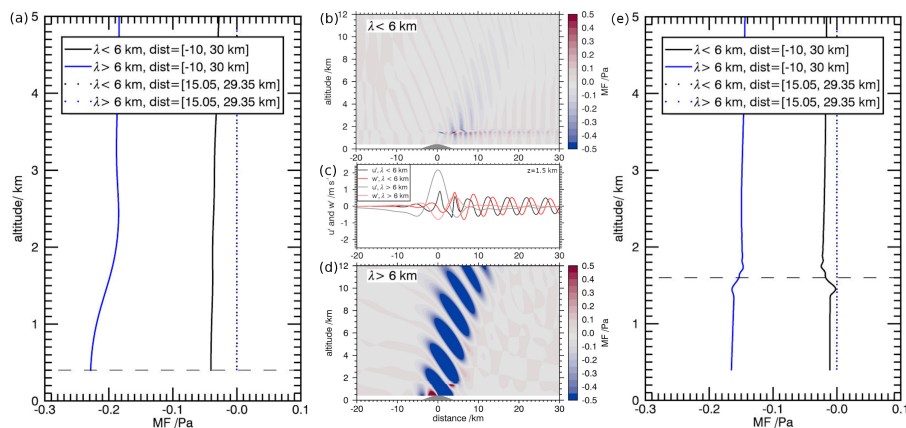

**Figure 13.** Momentum flux (profiles) for (a) RUN 1 without boundary layer inversion (Figs. 11a, 12a) and (b,d,e) Run 2 with boundary layer inversion (Figs. 11b, 12c) for two wave classes (horizontal wavelength smaller or larger 6 km). Profiles show averages for the full distance (solid) and for the downstream distance (dotted); horizontal dashed lines marks the top of the boundary layer. (c) shows $u'$ and $w'$ for the two wave classes at 1.5 km altitude revealing their phase relationship right below the inversion, i.e.- 90° for interfacial waves (black and red lines).

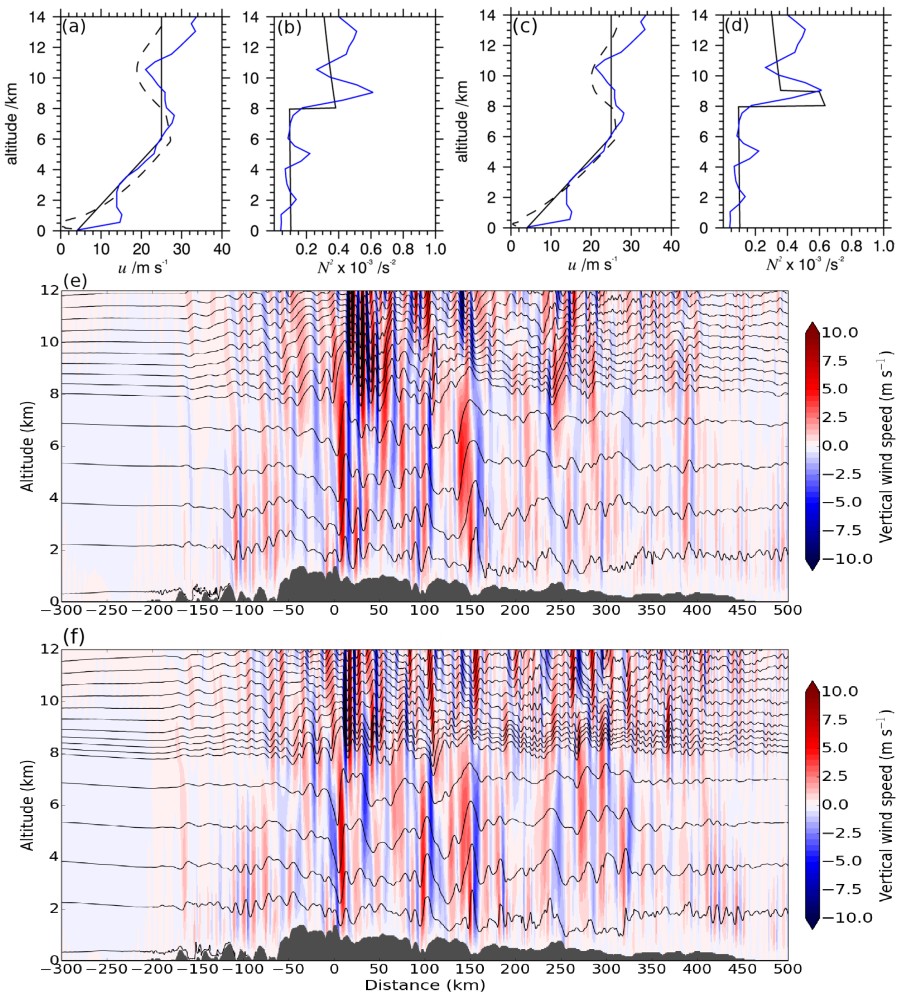

**Figure 14.** Initial profiles (black solid) and vertical velocity for the simulations with more realistic terrain without TIL (a, b, e; RUN 5) and with TIL (c, d, f; RUN 6). The initial profiles approximate the background conditions over southern Scandinavia on 28 January 2016 (blue profiles show the Stavanger radiosonde data). Negative shear above the tropopause establishes in the course of the simulations (a, d; black dashed, time = 16 h, distance = −150 km).

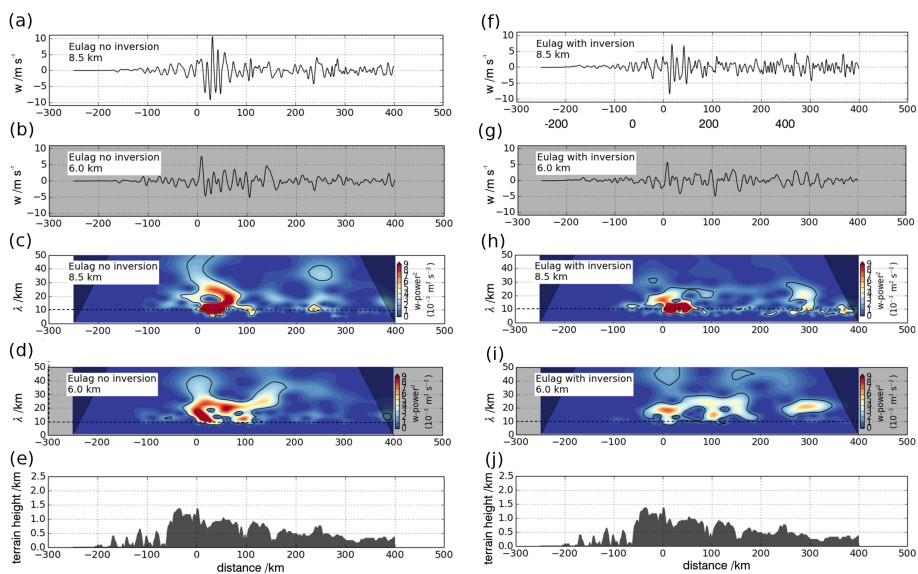

**Figure 15.** Vertical velocity and corresponding wavelet spectra of the simulations shown in Fig. 14 [left: without TIL (RUN 5), right: with TIL (RUN 6)]. a, c, f, h are at the altitude of the TIL and b, d, g, i below the tropopause. e and j show the terrain in the domain.

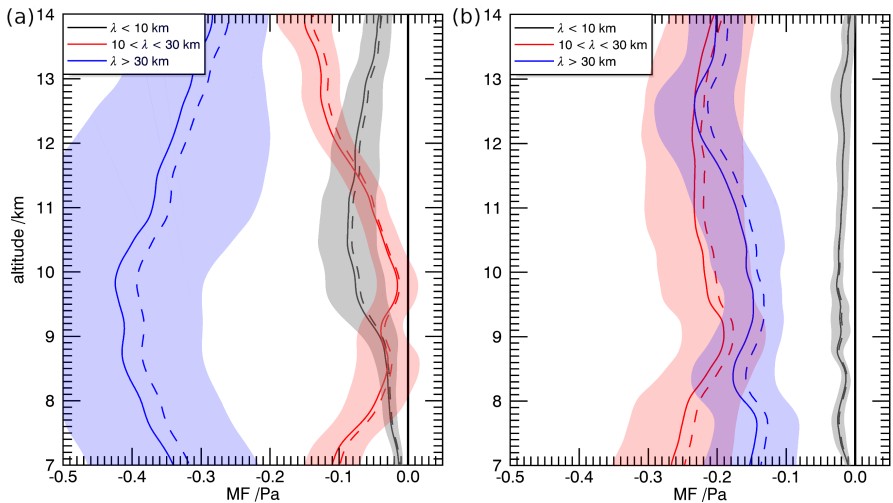

**Figure 16.** Averaged momentum fluxes for (a) RUN 5 (no TIL) and (b) RUN 6 (TIL) as mean for varying leg length (solid) with standard deviation (shading) and for the full leg distance [-300, 400 km] (dashed). Three wave classes are color coded and bold black line separates positive and negative fluxes

**Table 1.** Mountain width $L$, static stabilities of the lower and upper layer ($N_L$,$N_U$), strength of the inversion ($\Delta\theta$) and inversion height ($z_i$), upstream wind conditions, background density, and integration time of the EULAG simulations.

| RUN | $L$ /km | $N_L$ /s$^{-1}$ | $N_U$ /s$^{-1}$ | $\Delta\theta$ /K | $z_i$ /m | $U$ /m s$^{-1}$ | $\bar{\rho} = const.$ | time |
|---|---|---|---|---|---|---|---|---|
| 1 | 10 | 0.00 | 0.01 | 0.0 | 400 | 8 | yes | 96 min |
| 2 | 10 | 0.00 | 0.01 | 3.3 | 1600 | 8 | yes | 96 min |
| 3 | 5 | 0.00 | 0.02 | 6.6 | 1600 | 8 | yes | 190 min |
| 4 | 5 | 0.01 | 0.02 | 6.6 | 8000 | 8 | yes | 96 min |
| 5 | ASTER topo | 0.01 | 0.02 | 0.0 | 8000 | $4 \rightarrow 25$ | no | 16 h |
| 6 | ASTER topo | 0.01 | 0.02 | 20 | 8000 | $4 \rightarrow 25$ | no | 16 h |