# Peer review of "Airborne measurements and large-eddy simulations of small-scale Gravity Waves at the tropopause inversion layer over Scandinavia"

_Atmospheric Chemistry and Physics, 2020_

## Referee Comment (RC1) · Anonymous Referee #1 · 6 May 2020

[letterpaper, 12pt]article

1. **General Comments**

   This is an excellent paper that melds analysis of data from recent field campaigns with idealized numerical modelling to obtain insight into apparent interfacial waves in the tropopause inversion layer. I find it to be well written and enlightening from a scientific point of view, and it deserves publication in ACP. Nonetheless, I do have several (mostly minor) comments and questions.

2. **Specific Comments**

[Figure]

- In RF08 FL1, there is also a region of positive MF at $\lambda = 30\text{-}40$ km and $x = 200\text{-}250$ km that is almost equal in magnitude to the region of negative MF perturbations at $\lambda = 40\text{-}50$ km and $x = 120$ km. The latter region is attributed to vertically propagating waves in the text, but is there any explanation for the positive MF region?

- How long are the simulations run for? Are they at approximately steady-state?

- The initial profiles for the more realistic idealized simulations appear that they may be capable of supporting trapped waves due to the wind shear in the troposphere. There appear to be trapped-wave-like structures in the troposphere in both the no-TIL and TIL cases, which may also be leaking into the stratosphere, particularly in the TIL inversion case where there is an even larger Scorer parameter in the inversion. Where do these perturbations appear in the wavelet analysis? Could leakage of waves from the troposphere complicate the attribution of perturbations in the TIL to interfacial waves?

- When discussing waves with wavelength of $\sim 8$ km, it is very difficult to gain an appreciation of the structure with the aspect ratio in the vertical velocity plots. It would be nice to stretch out the horizontal axis a bit more for ease of interpretation.

- It might be nice to show one or two example soundings (with $\theta$) for the idealized cases outlined in Table 1, which would help illustrate the exact setup of the inversion.

- Line 45: The sentence beginning here is oddly worded. I think it would be better to make clear that "they" here refers to the fundamental characteristics of the hydrostatic approximation. At first reading it initially seems like "they" refers to the findings.

[Figure]

- Line 146: The $\rho_0$ of the Boussinesq approximation is not actually defined in the text.

- Line 175: It might be useful to note that the "critical horizontal wavelength" is also referred to as the Scorer parameter, especially since the Scorer parameter is referenced elsewhere in the text. It may also be helpful to have the equation for the Scorer parameter written in the paper as well.

- Line 194: I think it would be good to have a citation for alternating momentum fluxes being an indication of reflected and trapped waves.

- Line 280: "Interfacial small 280 scale waves are absent in the troposphere below the TIL (Fig. 15i) and in the case of no TIL (RUN 5, Fig. 14c and Fig. 15a, c)." For the no-TIL simulation, are you referencing below the would-be TIL or above or both? The referenced figures imply above, but text seems to imply either below or both.

3. **Technical Corrections**

- Line 304: Second word should be "were" instead of "where"
- Line 305: Extra "to" in "range of to"

---

## Referee Comment (RC2) · Anonymous Referee #2 · 12 May 2020

Summary:

This article presents an interesting observed case of trapped mountain waves that suggests the importance of interfacial dynamics along a tropopause inversion layer (TIL). The observations are unique and make the article a valuable contribution from the observational analysis alone. The study is further enhanced by interesting sensitivity studies using a 2D model comparing the wave response in both boundary layer inversion and TIL scenarios. I strongly recommend that the analysis done for Figure 13 be expanded to include the TIL simulation (see below). But overall the simulations represent a compelling initial investigation into this mechanism not previously observed

in the UTLS. Although the simulations lead to further questions, I agree that further investigation is appropriately left to future studies. The article is generally well-written, although a few points that should be rewritten (particularly in the Introduction) are highlighted below.

Specific comments:

paragraph starting at line 30: what is the point of this paragraph? I think you are trying to give an overview of our understanding of gravity wave propagation from the troposphere to the upper atmosphere, but you include a lot of older work just to say that it's more complicated than basic theory. That is not a new finding. The sentence at line 41 starting "Fine scale structures…" is directly relevant to your work. But the rest of the paragraph can be condensed/cut.

If you choose to keep more of the background information, several sentences are awkwardly worded or confusing and should be fixed:

1. line 40: "This makes the wave spectrum (i.e. wavelengths) being determined by the vertical varying wind and stability and not by the topography spectrum which affects the relative amplitudes": confusing sentence. Do you mean: "In other words, the wave spectrum (i.e. wavelengths) and wave amplitudes are determined by the vertical varying wind and stability and not by the topography spectrum."

2. lines 44-46: These two sentences are awkwardly stated and I'm not sure why they're needed.

line 82: a "tight" discussion? What do you mean by "tight" in this context?

line 115: compensating

line 118: need to state what alpha represents

Figure 6: Interesting figure, but the caption was hard to follow. A few rewrite suggestions: "Black contour lines mark regions significant at the 95%-confidence level. The

cone of influence is shaded in grey. Flight legs located below the tropopause (see labelled mean flight altitudes) are marked with grey background colour."

-also Figure 6: Is the black dashed line showing the thermal tropopause location?

line 238: "and a free slip lower boundary condition is used."

line 247: "The signal downstream of the terrain is"

Figure 13: This is a compelling figure for the boundary layer inversion (RUN 2) interfacial wave analysis, but why is this same analysis not shown for the TIL (RUN 4)? An additional panel would strengthen your argument.

―――――――――――――――――

---

## Author Comment (AC1) · 14 Jul 2020

**Authors' response to the comments of the anonymous Referee #1**

We appreciate the positive feedback and the valuable comments of the two anonymous reviewers #1 and #2 which we considered carefully for our revision. Thus, changes in the content of the manuscript arose, especially with respect to the momentum flux (figures and discussion). These changes became necessary due to findings we made when incorporating the reviewers' comments. We spent some more time on the momentum flux calculations and the assessment of their sensitivity and uncertainty and managed to find a way of analyzing the momentum fluxes in measurements and simulations such that they provide robust vertical profiles in terms of their trend. The range of uncertainty for the quantitative momentum flux values is also included. Previously, we were not aware of the sensitivity of the averaged momentum flux, however, also thanks to the reviewer comments, we consider the sensitivity of the momentum flux calculation on the background removal and leg length in the revised momentum flux analysis and in the discussion of the manuscript. Below you will find a detailed explanation of the changes in the manuscript as well as the point-to-point answers to the review comments. All modified figures and one new figure are also attached at the end of this document.

*Momentum flux*

In the framework of the revision of the paper manuscript, we tried to quantify the influence of the reflected and trapped waves in the troposphere on the momentum flux (MF) profile (and also derived the MF profiles for the realistic simulation runs 5 and 6). By doing so, we realized that the determination of the perturbations u' and w' was not properly or rather not extensively enough described in the previous version of the manuscript. Moreover, during this analysis it turned out that the resulting MF profile is sensitive with respect to the background fit and the length and start/end points of the flight leg (not only for the measurements but also for the most idealized case of interfacial waves on a boundary layer inversion). By separating the leg into 3 segments for the measurements, we had already shown that there is a clear variability. But only after doing more detailed analyses and sensitivity test, we found out that we had drawn incomplete conclusions for the MF profile of the interfacial waves. The positive MF in the vicinity of the TIL is actually not due to the interfacial waves but due to the longer waves (>30 km) that are influenced by the TIL. The leg-averaged MF of the interfacial waves is close to zero (a detailed explanation is given below). The former downstream segment 3 was longer than segments 1 and 2, and thus, longer waves remained in u' and w' of segment 3. Considering this, we have carefully revised the parts of the manuscript that deal with the MF profiles.

The most important points/changes are:

+ We use spectral filters to determine u' and w' for different wave classes (long, intermediate, and short waves, i.e. mainly mountain waves, reflected and trapped waves in the troposphere, and interfacial waves). We separate the wave classes based on their horizontal wavelengths as seen in

the wavelet power spectra. long waves > 6 km and short waves < 6 km for the boundary layer simulations (Run 1, 2). Long waves > 30 km, intermediate waves 10 km to 30 km, and short waves < 10 km for the measurements and the TIL/no-TIL simulations (Run 5, 6). The advantage of the spectral filter is that the wavelengths contained in u' and w' are clearly defined. For the usual background fit, the wavelengths vary with the length of the leg. This is not appropriate for the purpose of our analyses. A similar separation of wave classes can be found in Georgelin and Lott 2001.

additional reference:

[revised manuscript text omitted]

**1. General Comments**
This is an excellent paper that melds analysis of data from recent field campaigns with idealized numerical modelling to obtain insight into apparent interfacial waves in the tropopause inversion layer. I find it to be well written and enlightening from a scientific point of view, and it deserves publication in ACP. Nonetheless, I do have several (mostly minor) comments and questions.
We thank the reviewer for this positive feedback and his/her valuable comments and questions that helped to improve the manuscript.

**2. Specific Comments**
• In RF08 FL1, there is also a region of positive MF at λ = 30-40 km and x = 200-250 km that is almost equal in magnitude to the region of negative MF perturbations at λ = 40-50 km and x = 120 km. The latter region is attributed to vertically propagating waves in the text, but is there any explanation for the positive MF region?
Taking into account the new findings for the leg-averaged MF profile at different scales, we could add for the positive MF region:
…Positive MF for the long waves is found in RF08 FL1 at 220 km distance which could be caused by partial reflection of these waves at the TIL.
…
…
The most prominent feature is the kink reaching positive values for the long waves between 8-km and 9-km altitude. Negative fluxes of the same magnitude are found below. This strengthens the previous assumption that waves are partially reflected at the TIL.

• How long are the simulations run for? Are they at approximately steadystate?
Run1 and Run2: 96min, Run3: 190min, Run4: 96min
The initial disturbance created by the mountain during the initialization of the simulations has moved far enough downstream in the region of interest and the simulations have reached quasi steady state by that time.

Run5 &6: 977min (~16h) as it takes longer for the initial disturbance to reach the border of the 1008-km large domain. In contrast to the single mountain simulations, these simulations having more complex topography do not reach quasi steady state due to continuous interaction of waves from the different mountain peaks, tropospheric trapped waves, and interfacial waves.

We added this information to Sec. 3.2 and added an extra column in Table 1:

… The total integration time for these simulations is between 96 and 190 minutes. The initial disturbance created by the mountain during the initialization of the simulations has moved far enough downstream in the region of interest and the simulations have reached quasi steady state by that time. Table 1 summarizes the relevant initial parameters and total integration time for the different model runs.

… The total integration time for these simulations is 16~hours. This is longer than for the other set of simulations because it takes longer for the initial disturbance to reach the border of the larger domain.  In contrast to the single mountain simulations, these simulations having more complex topography do not reach quasi steady state due to continuous interaction of waves from the different mountain peaks, tropospheric trapped waves, and interfacial waves.

• The initial profiles for the more realistic idealized simulations appear that they may be capable of supporting trapped waves due to the wind shear in the troposphere. There appear to be trapped-wave-like structures in the troposphere in both the no-TIL and TIL cases, which may also be leaking into the stratosphere, particularly in the TIL inversion case where there is an even larger Scorer parameter in the inversion. Where do these perturbations appear in the wavelet analysis? Could leakage of waves from the troposphere complicate the attribution of perturbations in the TIL to interfacial waves?
These tropospheric trapped waves are seen in the wavelet analyses at scales of 10-30 km horizontal wavelengths (Fig. 6, Fig. 15). In the revised version, we demonstrate that it is possible to separate the interfacial waves from the tropospheric trapped waves based on their scales. We revised the analysis of the measurements and simulations and computed the MF for three different wave classes (> 30km, 10 km to 30 km, and < 10 km). The MF of the interfacial waves and of the reflected and trapped waves in the troposphere is similar and varies around zero (Fig. 10). Trapped waves in the troposphere are known to have leg-averaged MFs of around zero (Woods and Smith, 2010; Georgelin and Lott, 2001).

• When discussing waves with wavelength of ~ 8 km, it is very difficult to gain an appreciation of the structure with the aspect ratio in the vertical velocity plots. It would be nice to stretch out the horizontal axis a bit more for ease of interpretation.
We agree that the aspect ratio was not the best choice in the first draft of the manuscript. We changed the aspect ratio and alignment of the panels for Figs. 8, 9, and 14.

• It might be nice to show one or two example soundings (with θ) for the idealized cases outlined in Table 1, which would help illustrate the exact setup of the inversion.
Done, we added the four θ-profiles to Fig. 11.

• Line 45: The sentence beginning here is oddly worded. I think it would be better to make clear that "they" here refers to the fundamental characteristics of the hydrostatic approximation. At first reading it initially seems like "they" refers to the findings.
Done, we changed the sentence as suggested.

…The fundamental characteristics of the hydrostatic approximation are the absence of a mechanism which allows a wave to propagate horizontally and the consequent upward propagation of energy directly above the obstacle, regardless of the horizontal extent of the generating terrain…

• Line 146: The $\rho_0$ of the Boussinesq approximation is not actually defined in the text.
$\rho_0 = 1.225$ kg/m$^3$, we added this information to Sec. 2.3.

• Line 175: It might be useful to note that the "critical horizontal wavelength" is also referred to as the Scorer parameter, especially since the Scorer parameter is referenced elsewhere in the text. It may also be helpful to have the equation for the Scorer parameter written in the paper as well.
Done, we added the definition of the Scorer parameter to the introduction and the definition of the critical horizontal wavelength to Sec. 3.1.1.
…

$$\ell^2(z) = \frac{N^2(z)}{U^2(z)} - \frac{\partial^2 U(z)/\partial z^2 + \partial U(z)/\partial z / H}{U(z)} - \frac{1}{4H^2} \qquad (1)$$

…

The critical horizontal wavelength $(= 2\pi/\ell)$ …

• Line 194: I think it would be good to have a citation for alternating momentum fluxes being an indication of reflected and trapped waves.
We added the reference of Woods and Smith 2010 and updated Fig. 13 which now shows that alternating momentum fluxes indicate trapped waves due to the phase relationship of u' and w'. We added the cross-reference to the section in which Fig. 13 is described.
…
This alternating pattern is an indication for reflected and trapped waves (Woods and Smith 2010, see also Sec. 3.2).
…
• Line 280: "Interfacial small 280 scale waves are absent in the troposphere below the TIL (Fig. 15i) and in the case of no TIL (RUN 5, Fig. 14c and Fig. 15a, c)." For the no-TIL simulation, are you referencing below the would-be TIL or above or both? The referenced figures imply above, but text seems to imply either below or both.
We clarified the description accordingly.

…Small scale interfacial waves are absent in the case of no TIL (Fig. 14a, c). They are only found in the TIL (Fig. 15f, h) but not below (Fig. 15g, i)…

**3. Technical Corrections**
• Line 304: Second word should be "were" instead of "where"
Done.
• Line 305: Extra "to" in "range of to"
This paragraph was revised according to the results of the updated MF analyses.

… In the lower stratosphere, the leg-averaged MF was found to be positive (around 0.05 Pa). This is in contrast to the findings during DEEPWAVE where no positive leg-averaged MF was found in the lower stratosphere above New Zealand ([Fig. 5b in (Smith et al., 2016)]. However, analyses in Smith et al. (2016) are limited to waves having scales < 150 km and at least in ground-based lidar data

downward propagating waves were frequently observed in wintertime in the stratosphere above New Zealand (Kaifler et al., 2017).

additional reference:

Kaifler, N., Kaifler, B., Ehard, B., Gisinger, S., Dörnbrack, A., Rapp, M., Kivi, R., Kozlovsky, A., Lester, M., and Liley, B.: Observational indications of downward-propagating gravity waves in middle atmosphere lidar data, Journal of Atmospheric and Solar-Terrestrial Physics, 162, 16 – 27, https://doi.org/10.1016/j.jastp.2017.03.003, 2017.

**Modified and new figures**

- modified (new aspect ratio):

[Figure]

**Figure 8.** Vertical winds along flight leg RF07 FL2 measured by the DWL and in-situ instruments (marked by red horizontal lines) at flight level by the DLR Falcon.

- modified (new aspect ratio):

[Figure]

**Figure 9.** DWL measurements of (a) horizontal wind speed and (b) vertical wind speed and WALES measurements of (c) water vapour mixing ratio and (d) lidar reflectivity along flight leg RF08 FL1/PGS11 HL1 combined with corresponding in-situ measurements of HALO and DLR Falcon at flight level (marked by blue and red horizontal lines).

- modified (revised data analysis):

[Figure]

**Figure 10.** Leg-averaged momentum fluxes as mean for varying leg length (solid) with standard deviation (shading, error bars) along flight leg RF08 FL1 obtained from DWL (×) and in-situ measurements (●) which include also PGS11 HL1 and HL4. Three wave classes are colour coded and bold black line separates positive and negative fluxes.

- modified (additional panels):

[Figure]

**Figure 11.** Potential temperature and vertical velocity of the idealized simulations of the cases with (a) a neutral boundary layer without inversion and $N_U = 0.01 \text{ s}^{-1}$ (RUN 1), (b) a neutral boundary layer with an inversion of 3.3 K and $N_U = 0.01 \text{ s}^{-1}$ (RUN 2), (c) a neutral boundary layer with an inversion of 6.6 K and $N_U = 0.02 \text{ s}^{-1}$ (RUN 3), and (d) a stable troposphere ($N = 0.01 \text{ s}^{-1}$) with a TIL of 6.6 K and $N_U = 0.02 \text{ s}^{-1}$ (RUN 4).

- modified (revised data analysis, new panels):

[Figure]

**Figure 13.** Momentum flux (profiles) for (a) RUN 1 without boundary layer inversion (Figs. 11a, 12a) and (b,d,e) Run 2 with boundary layer inversion (Figs. 11b, 12c) for two wave classes (horizontal wavelength smaller or larger 6 km). Profiles show averages for the full distance (solid) and for the downstream distance (dotted); horizontal dashed lines marks the top of the boundary layer. (c) shows $u'$ and $w'$ for the two wave classes at 1.5 km altitude revealing their phase relationship right below the inversion, i.e.- 90° for interfacial waves (black and red lines).

- modified (new aspect ratio):

[Figure]

**Figure 14.** Initial profiles (black solid) and vertical velocity for the simulations with more realistic terrain without TIL (a, b, e; RUN 5) and with TIL (c, d, f; RUN 6). The initial profiles approximate the background conditions over southern Scandinavia on 28 January 2016 (blue profiles show the Stavanger radiosonde data). Negative shear above the tropopause establishes in the course of the simulations (a, d; black dashed, time $= 16$ h, distance $= -150$ km).

- new figure:

[Figure]

**Figure 16.** Averaged momentum fluxes for (a) RUN 5 (no TIL) and (b) RUN 6 (TIL) as mean for varying leg length (solid) with standard deviation (shading) and for the full leg distance [-300, 400 km] (dashed). Three wave classes are color coded and bold black line separates positive and negative fluxes

---

## Author Comment (AC2) · 14 Jul 2020

**Authors' response to the comments of the anonymous Referee #1**

We appreciate the positive feedback and the valuable comments of the two anonymous reviewers #1 and #2 which we considered carefully for our revision. Thus, changes in the content of the manuscript arose, especially with respect to the momentum flux (figures and discussion). These changes became necessary due to findings we made when incorporating the reviewers' comments. We spent some more time on the momentum flux calculations and the assessment of their sensitivity and uncertainty and managed to find a way of analyzing the momentum fluxes in measurements and simulations such that they provide robust vertical profiles in terms of their trend. The range of uncertainty for the quantitative momentum flux values is also included. Previously, we were not aware of the sensitivity of the averaged momentum flux, however, also thanks to the reviewer comments, we consider the sensitivity of the momentum flux calculation on the background removal and leg length in the revised momentum flux analysis and in the discussion of the manuscript. Below you will find a detailed explanation of the changes in the manuscript as well as the point-to-point answers to the review comments. All modified figures and one new figure are also attached at the end of this document.

*Momentum flux*

In the framework of the revision of the paper manuscript, we tried to quantify the influence of the reflected and trapped waves in the troposphere on the momentum flux (MF) profile (and also derived the MF profiles for the realistic simulation runs 5 and 6). By doing so, we realized that the determination of the perturbations u' and w' was not properly or rather not extensively enough described in the previous version of the manuscript. Moreover, during this analysis it turned out that the resulting MF profile is sensitive with respect to the background fit and the length and start/end points of the flight leg (not only for the measurements but also for the most idealized case of interfacial waves on a boundary layer inversion). By separating the leg into 3 segments for the measurements, we had already shown that there is a clear variability. But only after doing more detailed analyses and sensitivity test, we found out that we had drawn incomplete conclusions for the MF profile of the interfacial waves. The positive MF in the vicinity of the TIL is actually not due to the interfacial waves but due to the longer waves (>30 km) that are influenced by the TIL. The leg-averaged MF of the interfacial waves is close to zero (a detailed explanation is given below). The former downstream segment 3 was longer than segments 1 and 2, and thus, longer waves remained in u' and w' of segment 3. Considering this, we have carefully revised the parts of the manuscript that deal with the MF profiles.

The most important points/changes are:

+ We use spectral filters to determine u' and w' for different wave classes (long, intermediate, and short waves, i.e. mainly mountain waves, reflected and trapped waves in the troposphere, and interfacial waves). We separate the wave classes based on their horizontal wavelengths as seen in

the wavelet power spectra. long waves > 6 km and short waves < 6 km for the boundary layer simulations (Run 1, 2). Long waves > 30 km, intermediate waves 10 km to 30 km, and short waves < 10 km for the measurements and the TIL/no-TIL simulations (Run 5, 6). The advantage of the spectral filter is that the wavelengths contained in u' and w' are clearly defined. For the usual background fit, the wavelengths vary with the length of the leg. This is not appropriate for the purpose of our analyses. A similar separation of wave classes can be found in Georgelin and Lott 2001.

additional reference:

[revised manuscript text omitted]

**Summary:**
This article presents an interesting observed case of trapped mountain waves that suggests the importance of interfacial dynamics along a tropopause inversion layer (TIL). The observations are unique and make the article a valuable contribution from the observational analysis alone. The study is further enhanced by interesting sensitivity studies using a 2D model comparing the wave response in both boundary layer inversion and TIL scenarios. I strongly recommend that the analysis done for Figure 13 be expanded to include the TIL simulation (see below). But overall the simulations represent a compelling initial investigation into this mechanism not previously observed in the UTLS. Although the simulations lead to further questions, I agree that further investigation is appropriately left to future studies. The article is generally well-written, although a few points that should be rewritten (particularly in the Introduction) are highlighted below.
We thank the reviewer for this positive feedback and his/her valuable suggestions that helped to improve the manuscript. We revised and shortened the introduction as suggested. As already described above, we also revised the MF analysis and determined MF profiles for the no-TIL/TIL simulation with realistic topography.

**Specific comments:**
paragraph starting at line 30: what is the point of this paragraph? I think you are trying to give an overview of our understanding of gravity wave propagation from the troposphere to the upper atmosphere, but you include a lot of older work just to say that it's more complicated than basic theory. That is not a new finding. The sentence at line 41 starting "Fine scale structures…" is directly relevant to your work. But the rest of the paragraph can be condensed/cut.

If you choose to keep more of the background information, several sentences are awkwardly worded or confusing and should be fixed:

1. line 40: "This makes the wave spectrum (i.e. wavelengths) being determined by the vertical varying wind and stability and not by the topography spectrum which affects the relative amplitudes": confusing sentence. Do you mean: "In other words, the wave spectrum (i.e. wavelengths) and wave amplitudes are determined by the vertical varying wind and stability and not by the topography spectrum."

We decided to keep some of the background information for completeness but shortened the paragraph and revised the wording.

…

> 30    Vadas et al., 2003). GWs are propagating from their sources in the troposphere and the tropopause region (Sato et al., 2009; Fritts et al., 2016). However, the atmospheric temperature and wind structures influence the propagation of GWs and alter their properties.
>
>      Starting with the work of Queney (1948) and Scorer (1949), mountain wave (MW) propagation in the atmosphere was intensively investigated using theoretical and numerical methods. An important and well known result of these investigations is
> 35    that the stratospheric solution in a model taking into account a vertically varying background is not dominated by the classical solution of Queney (1948) but by reflected and downstream propagating (trapped) waves in the troposphere (Wurtele et al., 1987; Keller, 1994). The wave spectrum (i.e. wavelengths) is determined by the vertical varying wind and stability and not by the topography spectrum. The topography affects the relative amplitudes (Keller, 1994; Ralph et al., 1997). Fine scale structures in the atmosphere, such as sharp temperature inversions at the top of the boundary layer (Vosper, 2004; Sachsperger

…

2. lines 44-46: These two sentences are awkwardly stated and I'm not sure why they're needed.

We added a reference to mountain wave drag parametrizations to highlight that the hydrostatic approximation is still of importance nowadays.

…

Linear nonrotating hydrostatic wave theory is most commonly used by mountain wave parameterizations in weather and climate models to propagate these waves away from the subgrid-scale orography to higher levels (Eckermann et al., 2015).

line 82: a "tight" discussion? What do you mean by "tight" in this context?

We revised the sentence.

…

The results are discussed in Section 4 and Section 5 concludes the paper.

line 118: need to state what alpha represents

The sign was the "proportional to" sign. To avoid this misunderstanding we now give the exact definition.

…

leg-averaged momentum flux ($MF = \overline{\bar{\rho} u' w'}$)

Figure 6: Interesting figure, but the caption was hard to follow. A few rewrite suggestions: "Black contour lines mark regions significant at the 95%-confidence level. The cone of influence is shaded in grey. Flight legs located below the tropopause (see labelled mean flight altitudes) are marked with grey background colour."

-also Figure 6: Is the black dashed line showing the thermal tropopause location?

We clarified the caption as suggested. The black dashed line marks the 10-km horizontal wavelength.

**Figure 6.** Cross mountain flight legs of Falcon and HALO for (a) in-situ vertical wind and topography and (b) corresponding wavelets with the horizontal wavelength given on the y-axis. The dashed horizonal line marks 10-km horizontal wavelength that separates short and intermediate scale waves. Black contour lines mark regions significant at the 95%-confidence level. The cone of influence is shaded in grey. Flight legs located below the tropopause (see labelled mean flight altitudes) are marked with grey background colour. Time $t_0$ indicates when the aircraft was located at $X_0$ (see Fig. 1) and shows that PGS11 HL1 and RF08 FL1 (labelled with yellow boxes) took place nearly at the same time (HALO was flying 30 seconds behind Falcon).

line 238: "and a free slip lower boundary condition is used."
Done, we changed the sentence as suggested.

line 247: "The signal downstream of the terrain is"
Done, we changed the sentence as suggested.

Figure 13: This is a compelling figure for the boundary layer inversion (RUN 2) interfacial wave analysis, but why is this same analysis not shown for the TIL (RUN 4)? An additional panel would strengthen your argument.
As described above, we revised the MF analysis. Instead of showing the MF analysis for RUN 4 (which is in the Bousinesq framework), we did the analysis for RUNs 5 and 6. We added the MF profiles for these no-TIL/TIL simulation with realistic topography in new Fig. 16.
…
The pronounced kink in the MF profiles of the long waves (>30 km) and the short waves (<10km) in the altitude range of 7 km to 9 km is a clear feature of the effect of the TIL and not visible in the no-TIL simulation (Fig. 16a).
…

**Modified and new figures**

- modified (new aspect ratio):

[Figure]

**Figure 8.** Vertical winds along flight leg RF07 FL2 measured by the DWL and in-situ instruments (marked by red horizontal lines) at flight level by the DLR Falcon.

- modified (new aspect ratio):

[Figure]

**Figure 9.** DWL measurements of (a) horizontal wind speed and (b) vertical wind speed and WALES measurements of (c) water vapour mixing ratio and (d) lidar reflectivity along flight leg RF08 FL1/PGS11 HL1 combined with corresponding in-situ measurements of HALO and DLR Falcon at flight level (marked by blue and red horizontal lines).

- modified (revised data analysis):

[Figure]

**Figure 10.** Leg-averaged momentum fluxes as mean for varying leg length (solid) with standard deviation (shading, error bars) along flight leg RF08 FL1 obtained from DWL (×) and in-situ measurements (●) which include also PGS11 HL1 and HL4. Three wave classes are colour coded and bold black line separates positive and negative fluxes.

- modified (additional panels):

[Figure]

**Figure 11.** Potential temperature and vertical velocity of the idealized simulations of the cases with (a) a neutral boundary layer without inversion and $N_U = 0.01 \text{ s}^{-1}$ (RUN 1), (b) a neutral boundary layer with an inversion of 3.3 K and $N_U = 0.01 \text{ s}^{-1}$ (RUN 2), (c) a neutral boundary layer with an inversion of 6.6 K and $N_U = 0.02 \text{ s}^{-1}$ (RUN 3), and (d) a stable troposphere ($N = 0.01 \text{ s}^{-1}$) with a TIL of 6.6 K and $N_U = 0.02 \text{ s}^{-1}$ (RUN 4).

- modified (revised data analysis, new panels):

[Figure]

**Figure 13.** Momentum flux (profiles) for (a) RUN 1 without boundary layer inversion (Figs. 11a, 12a) and (b,d,e) Run 2 with boundary layer inversion (Figs. 11b, 12c) for two wave classes (horizontal wavelength smaller or larger 6 km). Profiles show averages for the full distance (solid) and for the downstream distance (dotted); horizontal dashed lines marks the top of the boundary layer. (c) shows $u'$ and $w'$ for the two wave classes at 1.5 km altitude revealing their phase relationship right below the inversion, i.e.- 90° for interfacial waves (black and red lines).

- modified (new aspect ratio):

[Figure]

**Figure 14.** Initial profiles (black solid) and vertical velocity for the simulations with more realistic terrain without TIL (a, b, e; RUN 5) and with TIL (c, d, f; RUN 6). The initial profiles approximate the background conditions over southern Scandinavia on 28 January 2016 (blue profiles show the Stavanger radiosonde data). Negative shear above the tropopause establishes in the course of the simulations (a, d; black dashed, time = 16 h, distance = −150 km).

- new figure:

[Figure]

**Figure 16.** Averaged momentum fluxes for (a) RUN 5 (no TIL) and (b) RUN 6 (TIL) as mean for varying leg length (solid) with standard deviation (shading) and for the full leg distance [-300, 400 km] (dashed). Three wave classes are color coded and bold black line separates positive and negative fluxes

---

## Author Comment (AC3) · 14 Jul 2020

In the supplement (pdf) of the response to the comments to anonymous Referee #2, it says "Authors' response to the comments of the anonymous Referee #1" instead of "#2". This is a typo. The pdf indeed contains the response to the comments of Referee #2. Please be aware, that both pdfs #1 and #2 start with the same preamble/introduction.